# The precursor of PI(3,4,5)P$_3$ alleviates aging by activating *daf-18*(*Pten*) and independent of *daf-16*

Dawei Shi[1,2,3,8], Xian Xia [1,2,3,8], Aoyuan Cui[3,4,8], Zhongxiang Xiong[1,5], Yizhen Yan [1,3], Jing Luo[1,6], Guoyu Chen[1,3], Yingying Zeng[1,5], Donghong Cai[1,3], Lei Hou [1,3], Joseph McDermott[1], Yu Li [4], Hong Zhang[3,7] & Jing-Dong J. Han [1,2✉]

Aging is characterized by the loss of homeostasis and the general decline of physiological functions, accompanied by various degenerative diseases and increased rates of mortality. Aging targeting small molecule screens have been performed many times, however, few have focused on endogenous metabolic intermediates—metabolites. Here, using *C. elegans* lifespan assays, we conducted a worm metabolite screen and identified an eukaryotes conserved metabolite, *myo*-inositol (MI), to extend lifespan, increase mobility and reduce fat content. Genetic analysis of enzymes in MI metabolic pathway suggest that MI alleviates aging through its derivative PI(4,5)P$_2$. MI and PI(4,5)P$_2$ are precursors of PI(3,4,5)P$_3$, which is negatively related to longevity. The longevity effect of MI is dependent on the tumor suppressor gene, *daf-18* (homologous to mouse *Pten*), independent of its classical pathway downstream genes, *akt* or *daf-16*. Furthermore, we found MI effects on aging and lifespan act through mitophagy regulator *PTEN induced kinase-1* (*pink-1*) and mitophagy. MI's anti-aging effect is also conserved in mouse, indicating a conserved mechanism in mammals.

[1] Key Laboratory of Computational Biology, Chinese Academy of Sciences–Max Planck Partner Institute for Computational Biology (PICB), Shanghai Institute of Nutrition and Health (SINH), Chinese Academy of Sciences Center for Excellence in Molecular Cell Science, Collaborative Innovation Center for Genetics and Developmental Biology, Shanghai Institutes for Biological Sciences (SIBS), Chinese Academy of Sciences (CAS), Shanghai 200031, P.R. China. [2] Peking-Tsinghua Center for Life Sciences, Academy for Advanced Interdisciplinary Studies, Center for Quantitative Biology (CQB), Peking University, Beijing 100871, P.R. China. [3] University of Chinese Academy of Sciences, Beijing 100049, P.R. China. [4] CAS Key Laboratory of Nutrition, Metabolism and Food Safety, SINH, SIBS, CAS, Shanghai 200031, P.R. China. [5] School of Life Science and Technology, ShanghaiTech University, Shanghai 201210, P.R. China. [6] College of Biological Sciences and Technology, Beijing Forestry University, Beijing, P.R. China. [7] National Laboratory of Biomacromolecules, CAS Center for Excellence in Biomacromolecules, Institute of Biophysics, CAS, 100101 Beijing, P.R. China. [8] These authors contributed equally: Dawei Shi, Xian Xia, Aoyuan Cui. ✉email: jackie.han@pku.edu.cn

Molecules that promote healthy aging have been searched and studied because of their potential to target human aging[1]. The aging process is closely linked to metabolism[2]. Metabolism is composed of chemical reaction networks, which can be represented with metabolites as nodes and enzymes as edges[3]. Metabolites, such as NAD$^+$ and related metabolites[4–6], spermidine[7], pyruvate[8], N-acetylglucosamine (GlcNAc)[9], and $\alpha$-ketoglutarate[10], have been reported to alleviate aging in model organisms.

The number of metabolites varies in magnitudes across the four kingdoms: tens of thousands in plants, thousands in humans, and one thousand in worm[11–13]. To investigate the influence of available metabolites on *C. elegans* lifespan, we conducted a preliminary screen on worm 100 metabolites and found that an evolutionarily conserved metabolite, *myo*-inositol (MI), significantly increased lifespan across a large range of concentrations.

MI is an extensively studied metabolite[14–17], whose isomers or derivatives participate in many fundamental physiological processes[18]. Among its nine possible stereo-isomers, MI is the most abundant[18]. MI is a common ingredient in food[19], and it is also used as a food supplement to alleviate a variety of diseases[20], including diabetes[21] and polycystic ovary syndrome (PCOS)[22]. An isomer of MI, D-*chiro*-Inositol, which does not exist endogenously in either *C. elegans* or *Drosophila melanogaster*, has been reported to extend fly lifespan through unknown mechanisms and was speculated to act through *dFOXO*[13, 23]. Inositol in the form of MI itself has not been reported to influence aging.

There are scores of MI derivatives, classified into three groups, phosphatidylinositol phosphates (PIs), inositol phosphates (IPs), and glycosyl phosphatidylinositols (GPI)[24–26]. Among them, abnormally high levels of PI(3,4,5)P$_3$, a key lipid component in the insulin and insulin-like growth factor signaling pathway (IIS), is known to promote cancer and shorten lifespan through activation of AKT in mouse/mammals[27]. Conversely, another derivative, IP$_7$ is reported to inhibit AKT in mouse[28], but has not been reported to influence aging. Both PI(3,4,5)P$_3$ and IP$_7$ in aging were indirectly studied by perturbations of their synthases, i.e., loss of function (LOF) of worm Phosphatidylinositol 3-kinase (*age-1*)[29], mouse liver phosphatidylinositol-4,5-bisphosphate 3-kinase (*Pi3k*)[30] and LOF of inositol hexaphosphate kinase (*Ip6k1*) in mouse[28], respectively. It remains unknown whether or how the other derivatives influence aging.

Here we found that MI extended worm lifespan and promoted healthspan, in other words, prevented the age-related decline in physiological functions. Genetic dissection of the MI metabolic pathway suggests that MI alleviates aging through its derivative PI(4,5)P$_2$, despite MI and PI(4,5)P$_2$ being precursors of PI(3,4,5)P$_3$. Moreover, its effects are completely dependent on the well-known tumor suppressor gene, *daf-18* (phosphatase and tensin, *Pten* homologue), which has many functions besides being an enzyme in the MI metabolic pathway, and partially depends on its downstream mitophagy regulator *pink-1* rather than its classical downstream *daf-16*. The anti-aging effects of MI were recapitulated in mice at phenotypic and transcriptome levels, suggesting a conserved mechanism in mammals.

## Results

**MI alleviates aging in worm**. MI is a cyclohexanehexol (Fig. 1a) that is metabolized in some bacteria, most archaea, and all eukaryotes[18]. The major reactions in the MI pathway are conserved from worm, mouse to human. Here we define MI metabolic pathway as the reactions forming an unbroken chain with MI upstream or downstream as annotated by the three KEGG pathways: Inositol phosphate metabolism, Phosphatidylinositol signaling system and Glycosylphosphatidylinositol (GPI)-anchor biosynthesis.

Through a small molecule screen with endogenous metabolites, we found that various concentrations of MI, from 1 up to 700 mM, and optimally at 500 mM, could extend worm lifespan (Supplementary Fig. 2d). To avoid the strong osmotic stress associated with these concentrations, we performed further analysis to delineate its molecular pathways at 50 mM MI in downstream experiments, which extended worm mean lifespan by 14% ($p < 0.01$, Fig. 1b). In addition, as osmotic effects, which can be induced by chemicals that alter osmolyte balance, like sorbitol and are found to extend worm lifespan[31], we used 50 mM mannitol (M) as an osmotic control, because it cannot be further metabolized in worms.

To further validate the effect of MI on lifespan, we constructed a worm strain, over-expressing inositol 3 phosphate synthase (*inos-1*), the rate-limiting enzyme to synthesize MI in most species, including worms and mammals[32] as a fusion protein with GFP, and backcrossed to N2 six times. We found that this *inos-1::GFP* overexpression (OE) strain had a 19% increase in lifespan compared with GFP only OE strain ($p = 0.002$, Fig. 1c).

The second step in the MI synthesis pathway is catalyzed by the enzymes inositol monophosphatase (TTX-7) or Y6B3B.5, to produce MI from the three types of inositol monophosphate (IP). IP is produced by INOS-1 or the degradation product of inositol phosphates. The *ttx-7* LOF allele *nj51*, shortened worm lifespan by 8% ($p < 0.001$, Fig. 1d).

We further examined MI's effect on the transcriptome changes. As expected, MI induced a transcriptome change opposite to the changes induced by aging, in particular, aging downregulated genes were significantly upregulated by MI ($p < 0.0001$, Fig. 1e), and aging upregulated genes showed a trend towards downregulation by MI ($p = 0.1466$, Fig. 1f).

**MI promotes worm healthspan**. To examine whether MI also promotes healthspan, we first scored worm mobility by counting body-bending rate and pumping rate in adult day (AD) 4 to day 10. Both MI treatment and *inos-1* OE increased worm mobility (*$p < 0.05$ on AD_10 for MI, both AD_6 and AD_10 for *inos-1* OE, Fig. 2a, b), and alleviated worm pumping-rate declines during aging (*$p < 0.05$ on AD_6 for MI or AD_8 for *inos-1* OE, Supplementary Fig. 2a, b).

MI was reported to reduce body fat in mammals[33], and we found that MI also decreased worm fat storage after examining lipid oil red O (ORO) staining. In contrast, the osmotic control M increased worm body fat, compared to untreated control (***$p < 0.001$, Fig. 2c).

We also checked the effect of MI on worm reproduction and found that it did not obviously affect worm brood size when treating the worms from L4 stage to AD_8 (Supplementary Fig. 2c).

As MI supplementation enhanced worm mobility, we asked whether MI could rescue the phenotypes of worms modeling mitochondria-related disease. We found that MI but not M increased the mobility of Parkinson's disease (PD) model ($\alpha$-synuclein::YFP, *$p < 0.05$, Fig. 2d). Consistently, we found MI decreased the $\alpha$-SYNUCLEIN puncta in the PD model (*$p < 0.05$, Fig. 2e, f).

**MI promotes mouse healthspan**. Next, we examined whether the health effects of MI were conserved in mouse. We treated middle-aged female C57BL/6J mice with 580 mg MI per kg bodyweight by gavage, twice per week, from 9 months of age to 12 months, we also set up CTL, Pre-treatment CTL and Young CTL groups, with the latter two groups from other batches of mice than CTL and MI. All mice were tested and sacrificed at the same time (Fig. 3a). For the mouse experiment, we chose the lowest MI concentration

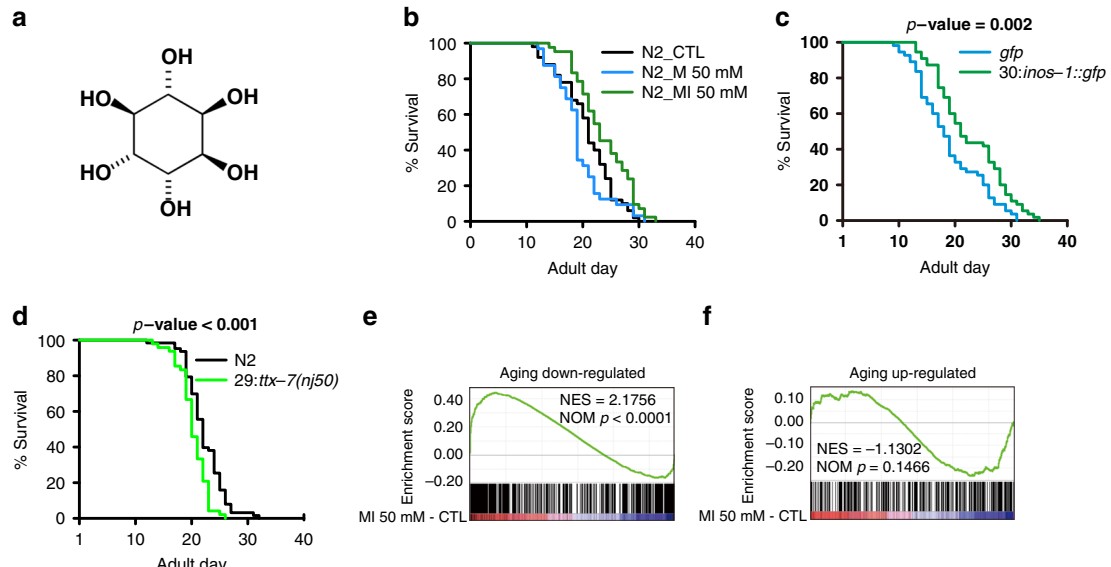

**Fig. 1 MI promotes worm longevity. a** Chemical structure of MI. **b** Survival curves of N2 worms treated with or without 50 mM M (log-rank test $p < 0.001$) or MI ($p < 0.01$), $N = 10$. **c** Survival curves of *inos-1* OE and control N2 worms (log-rank test $p = 0.002$). The number before ":" in a label indicates the reaction step in Fig. 4a, $N = 11$. **d** Survival curves of *ttx-7* LOF and control N2 worms (log rank test $p = 2.00E-04$), $N = 5$. **e**, **f** GSEA results show worm aging downregulated or upregulated gene expressions measured by RNA-seq FPKM $\log_2$-fold-change, were reversely regulated by 50 mM MI (GSEA $p < 0.0001$ for aging downregulated genes and $p = 0.1466$ for aging upregulated genes, respectively), $N = 1$. Source data are provided as a Source Data file and Supplementary Dataset 1.

reported in literature[33–35], 580 mg/kg. At this concentration, we did not observe any liver, kidney, or overall toxicity.

We first examined the transcriptome changes induced by MI in mouse muscle. MI treatment reversed the aging-related muscle transcriptome changes, as it did in the worm, i.e., it significantly increased the expression of aging downregulated genes ($p < 0.0001$, Fig. 3b), and showed a trend towards decreasing aging upregulated genes ($p = 0.0701$, Fig. 3c). The Pearson correlation coefficient (PCC) between changes induced by aging and by MI is $-0.509$, demonstrating a global anti-aging effect at the transcriptome level.

Phenotypically, the MI treated mice not only ran longer distance in open-field tests (*$p < 0.05$, ***$p < 0.001$, Fig. 3d), but also displayed more powerful muscle strength in grip-tests (***$p < 0.001$, Fig. 3e), MI was previously reported to decrease body fat[33], and here we found MI induced a trend towards reduced body fat, although it was insignificant in this cohort of mice ($p = 0.2$, Fig. 3f).

Based on all expressed genes (FPKM > 0 in any sample), PCA of all nine samples (three biological replicates for all Young_CTL, CTL, and MI) revealed that MI treated muscle samples were closer to CTL samples on PC1 and similar to Young_CTL samples on PC2 (Supplementary Fig. 3a, PC1 = 46%, PC2 = 27%). Interestingly, the effect of MI mostly showed on PC2, which is positively related to mitochondrial function (Supplementary Fig. 3a–b). The most significantly upregulated genes by MI treatment, as compared to CTL samples, include *Ddit4*, *Npas2*, *Arrdc3*, *Foxo1*; the most downregulated genes include *Cish*, *Zfp503* (Supplementary Fig. 3c).

We further examined the differentially expressed genes (FDR < 0.2) by clustering analysis. Cluster 0 and 1, related to cell adhesion and adaptive immunity, respectively, were downregulated genes comparing 12 vs 3-month mice, and were upregulated by MI; cluster 4 and 5 related to growth regulation and ion transport, respectively, were upregulated genes comparing 12 vs 3-month mice, and downregulated by MI. We also found MI induced downregulation of blood vessel development-related cluster 2 and upregulation of transcription related to cluster 3 genes, which were not drastically changed from M3 to M12 (Fig. 3g).

### The MI derivative PI(4,5)P2 is the effector for MI induced longevity

Next, we asked whether MI exerts its effects through its derivatives in the metabolic pathway. To address this, we examined the lifespan of worms with LOF of enzymes in all steps of the MI metabolic pathway, by using RNAi or mutants (Fig. 4a; Supplementary Fig. 4a–o, Supplementary Fig. 5, Supplementary Table 3). First, we looked into the inositol pyrophosphates of the MI derivatives and found that reduction of IP8 by knocking down its synthases, F46F11.1 (step 27) and F30A10.3 (step 28), decreased worm lifespan ($p < 0.02$, Supplementary Fig. 4m and $p = 0.08$, Supplementary Fig. 4n, respectively). IP7 was reported to be an Akt inhibitor in mouse[28], IP6 supplementation also extended worm lifespan in a dose-dependent manner (Supplementary Fig. 4p). Worms cannot produce IP7 from MI, due to the lack of an enzyme to synthesize IP6 from IP5 (Fig. 4a; Supplementary Fig. 1a, b). Therefore, MI's longevity effect is unlikely to be attributed to IP7.

Next, we looked through the PIs, and found that LOFs of *ttx-7* (step 29, $p < 0.001$, Fig. 2d), CDP-diacylglycerol-inositol 3-phosphatidyltransferase (*pisy-1*, step 1, $p < 0.001$, Fig. 4b), phosphatidylinositol 4-kinase B (*pifk-1*, step 9, $p = 0.121$, Figs. 4c) and 1-phosphatidylinositol-4-phosphate 5-kinase (*ppk-1*, step 13, $p < 0.001$. Supplementary Fig. 4g), which are enzymes in the flux to produce PI(4,5)P2 from IP monomers, all reduced worm lifespan.

In the MI metabolic pathway, PI(4,5)P2 exists as a hub. Six enzymes catalyze the reactions to or from PI(4,5)P2. They are 1-phosphatidylinositol-5-phosphate 4-kinase (*ppk-2*, step 7), phosphatidylinositol phospholipase C, beta (*egl-8*, step 18) and four other enzymes in two reversible reactions, i.e., *ppk-1* (forward, step 13) and synaptojanin (*unc-26*, reverse, step 14), *age-1* (forward, step 15) and *daf-18* (reverse, step 17). *ppk-2* has been

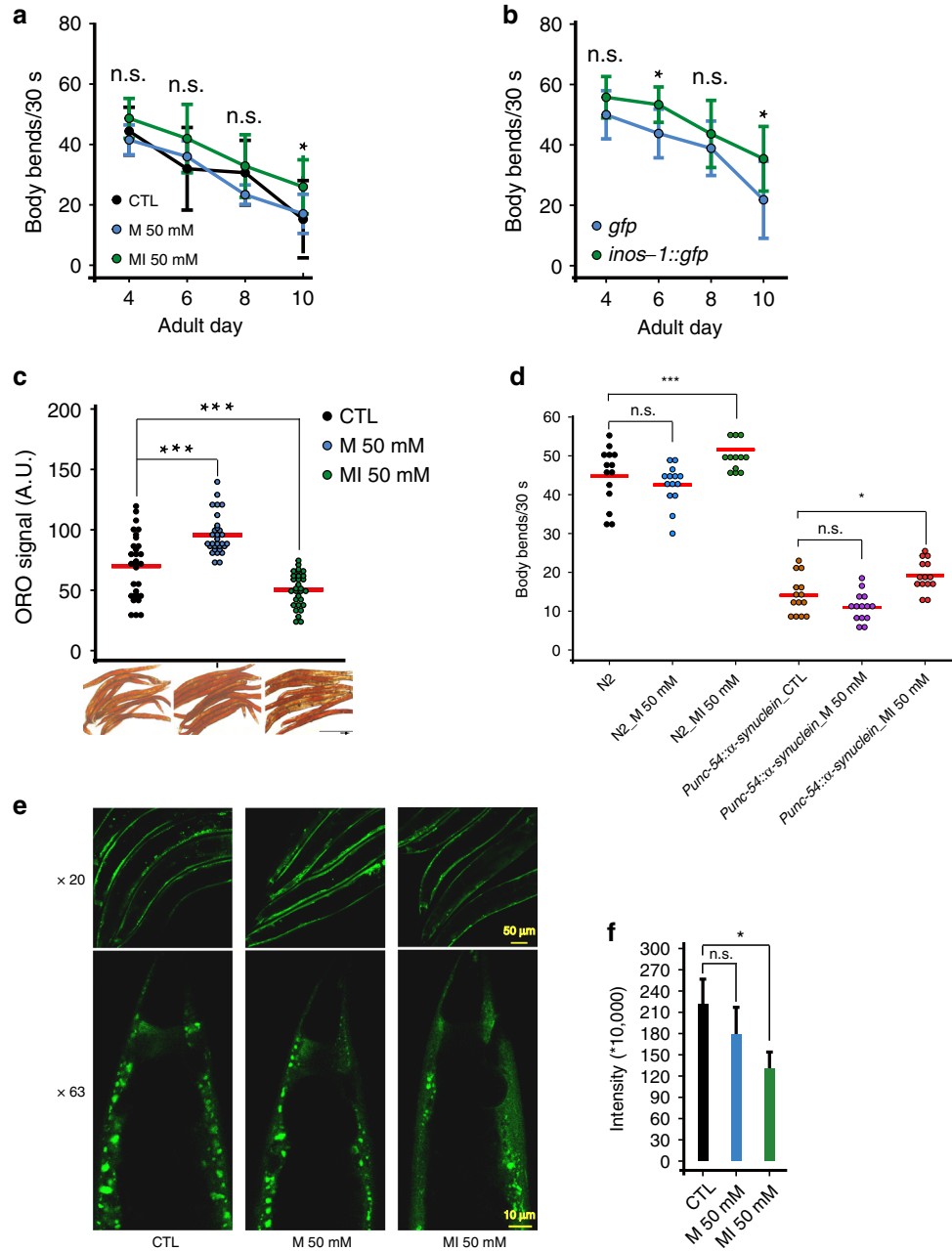

**Fig. 2 MI supplementation alleviates worm health decline during aging. a** 50 mM MI alleviated worm mobility decline from AD_4 to AD_10 (Data are presented as mean values ± SEM, $n = 10$, two-sided $t$-test *$p < 0.05$ on AD_10), $N = 2$. **b** $inos\text{-}1$ OE alleviated worm mobility decline (Data are presented as mean values ± SEM, $n = 10$, two-sided $t$-test *$p < 0.05$ on AD_6 and AD_10), $N = 2$. **c** ORO staining shows MI suppressed worm fat storage on AD_10 ($n = 30$, two-sided $t$-test ***$p < 0.001$), $N = 6$. **d** MI promoted the mobility of the worm model of PD ([$unc\text{-}54p\text{::}\alpha\text{-}synuclein\text{::}YFP + unc\text{-}119(+)$] ($n = 14$, two-sided $t$-test *$p < 0.05$), $N = 3$. **e** Representative images show MI supplementation decreased $\alpha$-SYNNUCLEIN::YFP aggregates in this PD model (20× labels images under 20x microscopy, scale bar 50 μm; 63× for 63x microscopy scale bar 10 μm). **f** Quantification of Fig. 2e like worms but on one worm tail region per image, under 63x CONFOCAL microscopy (Data are presented as mean values ± SEM, $n = 30$, two-sided $t$-test *$p < 0.05$), $N = 2$. Source data are provided as a Source Data file.

reported not to be a main PI(4,5)P$_2$ producer[36], although our data showed that knocking down $ppk\text{-}2$ by RNAi did not obviously influence worm lifespan ($p = 0.886$, Supplementary Fig. 4d), a $null$-allele $pk1343$ was reported to shorten worm lifespan[37]. For the other five enzymes, LOF of the two enzymes that produce PI(4,5)P$_2$, $ppk\text{-}1$ (step 13, $p < 0.001$. Supplementary Fig. 4g), and $daf\text{-}18$ (step 17, $p < 0.001$, Fig. 4d) shortened lifespan, and LOF of the three enzymes that consume PI(4,5)P$_2$, $unc\text{-}26$ (step 14), $age\text{-}1$ (step 15) and $egl\text{-}8$ (step 18) all extended lifespan ($p < 0.001$ for all three, Fig. 4d). All the three $egl\text{-}8$ LOF alleles, $nj77$, $n488$,

$md1971$ extended worm lifespan ($p < 0.001$ for all, Supplementary Fig. 4h, i).

Due to the instability when given as a supplement, and negative charge of PI(4,5)P$_2$, which makes it impermeable to plasma membranes, we could not test whether PI(4,5)P$_2$ supplementation promotes longevity.

We therefore tested heterozygous LOF of $ppk\text{-}1$ (as $ppk\text{-}1$-NL is lethal), the main enzyme to produce PI(4,5)P$_2$, shortened worm lifespan ($p < 0.001$, Fig. 4d), which was further confirmed by RNAi experiment ($p < 0.001$, Supplementary Fig. 4g). Furthermore,

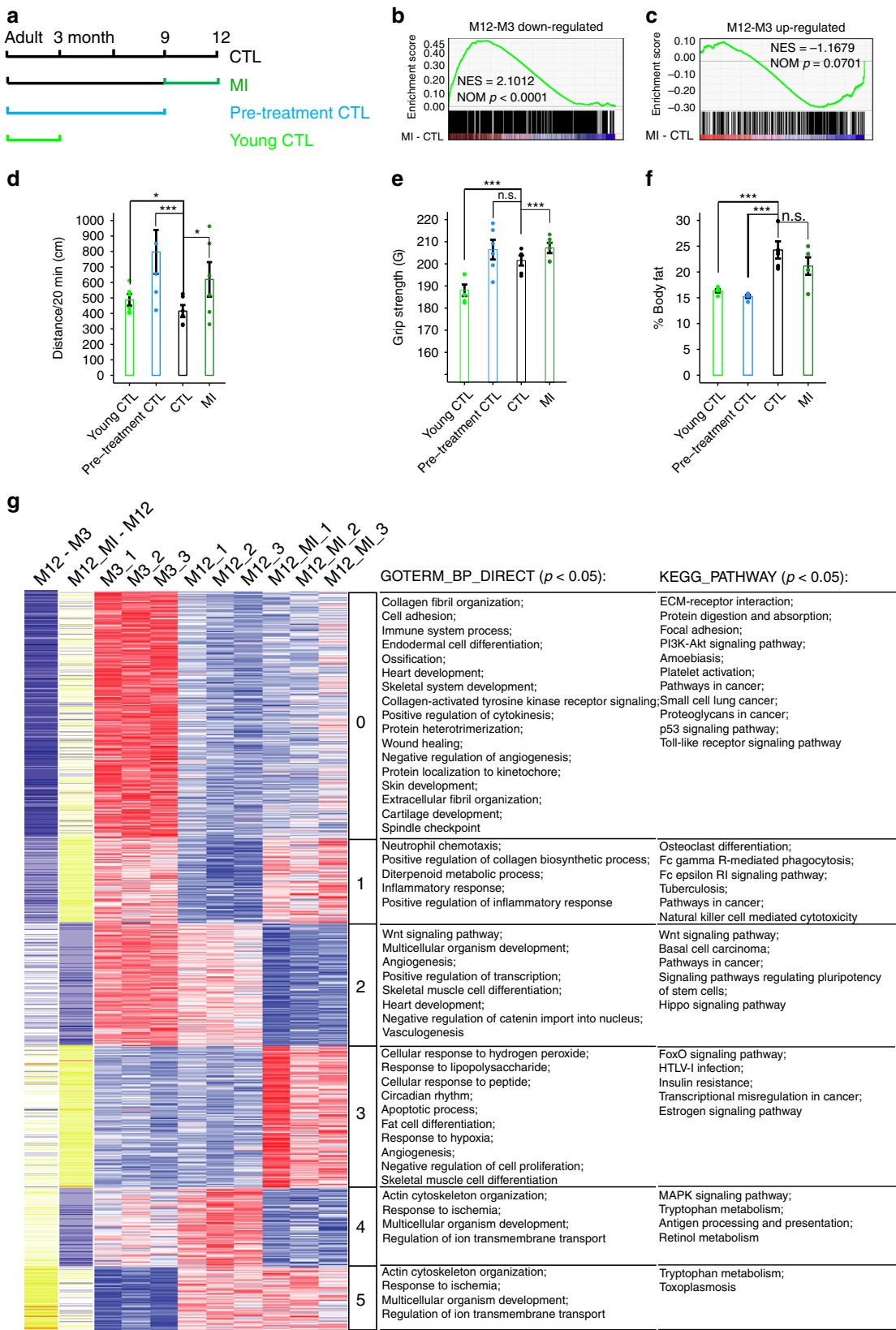

we found LOF of *ppk-1* also partially blocked MI's longevity effect (Fig. 4e). In addition, utilizing ELISA assays, we found a trend towards decreased PI(4,5)P$_2$ level in the worm aging process (albeit insignificant, $p = 0.197$, and could be reversed by MI treatment (**$p < 0.01$, Fig. 4f).

***daf-18* null-allele (NL) completely blocks the aging-alleviating effect of MI**. To identify the downstream target of MI or PI(4,5)P$_2$, we tested many genes in classical lifespan-regulatory signaling pathways for epistasis analysis, including *age-1, daf-16, rsks-1, ife-2, sir-2.1, glp-1, isp-1, aak-2, jnk-1, nhr-49*, and found that, none

**Fig. 3 MI promotes health in middle-aged mice. a** Design of mouse experiment. **b, c** GSEA results show mouse aging-downregulated or upregulated gene expressions, measured by RNA-seq $\log_2$-fold-change in FPKM, were reversely regulated by MI supplementary (GSEA $p < 0.001$ and $p = 0.0701$ respectively), $N = 1$. **d** MI increased mouse running distance in 20 min (Data are presented as mean values ± SEM, $n = 6$, two-sided $t$-test *$p < 0.05$, $p = 0.028$ between CTL and MI), $N = 2$. **e** MI increased mouse limbs grip strength (Data are presented as mean values ± SEM, $n = 6$, two-sided $t$-test *$p < 0.05$, $p = 0.0001$ between CTL and MI), $N = 2$. **f** MI decreased mouse body fat content (Data are presented as mean values ± SEM, $n = 8$ for CTL, $n = 4$ for others, two-sided $t$-test n.s. not significant, $p = 0.12$ between CTL and MI), $N = 1$. **g** Clustering of the aging-related genes, which were reversed by MI. Cluster 0 and 1 are aging downregulated genes, that were upregulated by MI, cluster 4 and 5 are aging upregulated genes, that were downregulated by MI. Cluster 2 or 3 show genes, which didn't change obviously from M3 to M12, but were downregulated or upregulated by MI, respectively. Source data are provided as a Source Data file and Supplementary Dataset 1.

of these mutants could largely block the lifespan effect of MI. Considering that $PI(4,5)P_2$ has been reported to bind and activate PTEN to dephosphorylate $PI(3,4,5)P_3$[38, 39], we tested a *PTEN* ortholog *null*-allele in the worm, *daf-18*-NL (*ok480*). We found that *ok480* could fully block MI lifespan extension (Fig. 5a) or *inos-1* OE (Fig. 5b) induced longevity. In addition, the anti-aging effect of MI on the transcriptome was also blocked by *daf-18*-NL ($p < 0.001$ in both directions, Fig. 5c, d). *daf-18*-NL also blocked the mobility promoting (Fig. 5e) and fat-reducing effects of MI (Fig. 5f). Next, we asked whether MI supplementation influences *daf-18 (Pten)* mRNA, protein status or activity. Since there was not a good antibody to detect worm DAF-18 protein, we examined PTEN levels with or without MI supplementation, in several tissues of mice sacrificed after health parameter assays. We found that mouse PTEN protein showed a trend towards upregulation in the muscle ($t$-test $p = 0.1$, Fig. 5g, h). *PTEN* is one of the most frequently mutated tumor suppressor genes in humans. Notably, *PTEN* is conserved from worm to human, human *PTEN* could rescue *daf-18*-NL worm[40, 41], suggesting canonical orthology between the two genes.

**PINK-1/PINK1 and mitophagy mediate MI effects.** As a lipid phosphatase, DAF-18/PTEN dephosphorylates $PI(3,4,5)P_3$, which activates AKT, which subsequently inhibits DAF-16/FOXO. However, *akt-1(ok525)* and *akt-2(ok393)* (Supplementary Fig. 6a, b) or a *daf-16 null*-allele (*mgDf50*) (Fig. 6a) could not block the longevity effect of MI supplementation. This suggested that the longevity effect of MI does not act through enhancing PTEN's activity to dephosphorylate $PI(3,4,5)P_3$, and the binding of $PI(4,5)P_2$ to PTEN might be essential in other functions of PTEN.

As mitophagy plays a central role in the PD model and a mammalian PTEN isoform was recently reported to enhance mitophagy[42–45], we examined whether the mitophagy regulator, *pink-1* mediates the longevity effect of MI[46]. We found *pink-1*-NL (*ok3538*) mostly, though not completely, blocked the longevity phenotype (Fig. 6b) and the mobility phenotype of MI ($p < 0.05$, Fig. 6c). LOF with *pink-1*'s target gene, the ortholog of human *PARKIN*, *pdr-1*-NL (*gk448*) partially blocked MI induced lifespan extension (Supplementary Fig. 6c). We further tested in worm mobility assay two mitochondrial fission genes, *drp-1* and *fis-2*, which are necessary for the mitophagy process[47]. Both could block MI's mobility enhancement effect (Supplementary Fig. 6d). These prompted us to investigate whether MI could promote mitophagy at the molecular level. We found that PINK1, rather than PARKIN protein in differentiated mouse muscle C2C12 cells, was upregulated by MI supplementation (**$p < 0.01$, Fig. 6d, e). Consistently, using aggregates of the *C. elegans* p62 homolog SQST-1 (SQST-1::GFP) as an indicator of autophagic flux[48], we found MI decreased SQST-1::GFP puncta in worm muscle, which indicates the enhancement of autophagy (*$p < 0.05$, Fig. 6f, g). Furthermore, with another autophagy marker GFP::lgg-1 (*C. elegans* homolog of mammalian *Atg8*) transgenic worms, MI promoted the co-localization GFP:: LGG-1 and mitotracker, indicating enhanced mitophagy were induced by MI treatment

(Fig. 6h, i). As mitophagy is known to promote mitochondria homeostasis and activity, we further examined whether MI increased mitochondria oxygen consumption rate (OCR) of mitochondria in both worms and differentiated C2C12 cells, which we observed with regard to maximal respiratory capacity (***$p < 0.001$ for basal OCR and was not significant for maximal OCR in the worm, respectively, Fig. 6j, k). At the transcript level, we examined mitochondria-related genes in the mouse muscle RNA-seq data and found that age-dependent changes therein were largely reversed by MI supplementation (PCC = −0.313, Supplementary Fig. 6e). These results demonstrate MI, an endogenous metabolite, promotes mitophagy.

## Discussion

MI is an evolutionarily ancient metabolite, existing in all eukaryotic organisms, and is metabolized in a highly conserved metabolic pathway. Here we identified it has an anti-aging effect, that acted through its conversion to $PI(4,5)P_2$ and downstream activation of PTEN and mitophagy, and not through the classical PTEN downstream insulin IGF-1 pathway components, namely, AKT and DAF-16 (Fig. 6l).

We considered whether the observed MI effects were influenced by hyperosmotic stress. Given worm has specific transporters to transfer MI but not the control M across the plasma membrane[49], the osmotic pressure would differ between the two at an equal concentration. However, both did not result in strong osmotic effect at 50 mM, and when we compared their effect in RNA level at 500 mM, the RNA-seq based differential gene expression analysis showed M induced stronger alteration of osmotic stress-related genes, especially the osmotic marker gene *gpdh-1*[50] (Supplementary Fig. 6f). In addition, *gpdh-1*-LOF cannot block MI's lifespan extension effect (Supplementary Fig. 6g). Thus, in our studies using 50 mM MI, changes in osmolarity should not be relevant, which is consistent with our results using *inos-1* OE worms and mutants in the MI metabolic pathway.

FUdR interferes with lifespan assay in many cases[51], and in our study, it also diminished the effect of MI. A possible cause is that, although worms are a post-mitotic animal, which does not need to replicate nuclear DNA, they still need to synthesize mitochondrial DNA to generate new mitochondria, and therefore mitochondria-dependent effects could be blocked by FUdR.

We also found that lifespan effects differed between live and killed OP50, although MI is not a metabolite utilizable by *E. coli*. We suspect that there might be some indirect effects of live bacteria to affect MI concentrations and render it less effective.

MI is often consumed at a large amount, without obvious adverse effects[52]. We found that up to 700 mM MI still extended the lifespan of worms, with a peak of 36% increased mean lifespan at 500 mM (Supplementary Fig. 2d). The longevity effect at both 50 and 500 mM as well as that by *inos-1* OE can be completely blocked by *daf-18*-NL (Supplementary Fig. 6h).

Although aberrant $PI(3,4,5)P_3$ elevation shortens lifespan, our results suggest that decreasing its precursor, $PI(4,5)P_2$, also shortens lifespan. Abnormal $PI(3,4,5)P_3$ elevation is a feature of

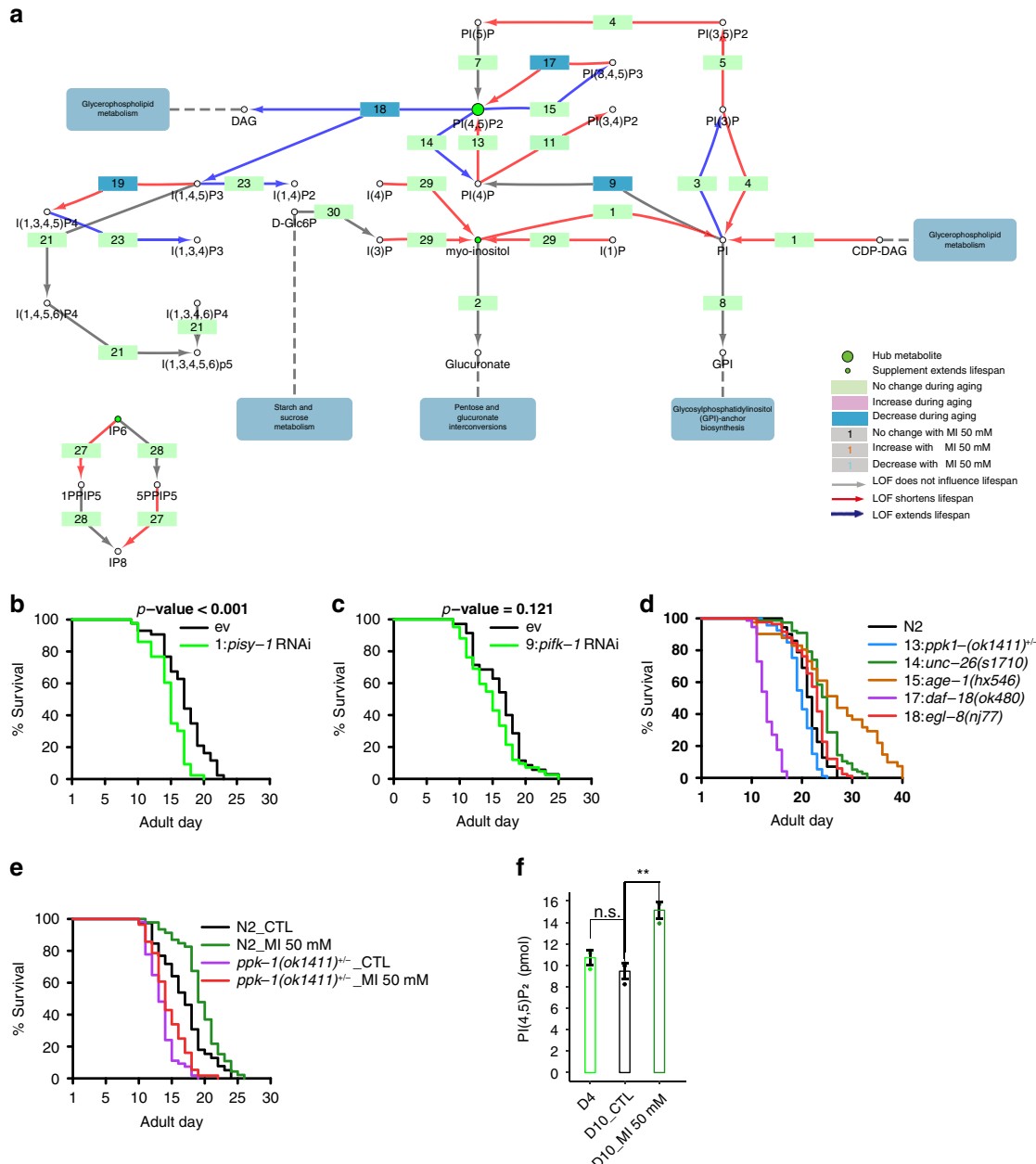

**Fig. 4 PI(4,5)P2 is the effector of MI. a** MI metabolic pathway in *C. elegans*. The arrowhead shows the reaction direction. The number labels the reaction step, which can be catalyzed by one or more enzymes. Red arrows indicate LOF of the enzyme shortened lifespan, cyan is the opposite with only log-rank test $p < 0.05$ results labeled. The color of the enzyme node shows the mRNA increased (plum) or decreased (blue) with age. The label color of the reaction number indicates MI supplementation increased (purple) or decreased (cyan) the enzymes' transcripts. Only results with $\log_2$ fold change > 0.26 are labeled. **b**, **c** Survival curves of empty vector control, *pisy-1* RNAi (log-rank test $p = 4.30E\text{-}05$, $N = 2$), and *pifk-1* RNAi worms (log-rank test $p = 0.121$, $N = 3$), respectively. **d** Survival curves of control and mutant worms with LOF enzymes producing or degrading PI(4,5)P2. (13:*ppk-1(ok1411)*+/−, log-rank test $p = 1.02E\text{-}04$, $N = 5$; 14:*unc-26(s1710)*, log-rank test $p = 2.23E\text{-}08$, $N = 6$; 15:*age-1(hx546)*, log-rank test $p = 5.41E\text{-}08$, N = 4; 17:*daf-18(ok480)*, log-rank test $p = 1.54E\text{-}35$, N = 8; 18:*egl-8(nj77)*, log-rank test $p = 3.33E\text{-}11$), $N = 6$. **e** *ppk-1* heterozygous mutant partially blocked MI longevity effect, $N = 2$. **f** ELISA results show PI(4,5)P2 level decreased during worm aging (Data are presented as mean values ± SEM, n.s. $p = 0.20$), and was reversed by MI supplement on AD_10 (two-sided *t*-test **$p = 0.003$, $N = 3$. Source data are provided as a Source Data file.

cancers and many other diseases, while aberrant $PI(3,4,5)P_3$ decline results in diabetes. Therefore, a balance between $PI(4,5)P_2$ and $PI(3,4,5)P_3$ appears more vital than simply inhibiting $PI(3,4,5)_3$, i.e., controlling $PI(3,4,5)_3$ level should not be achieved by blocking $PI(4,5)P_2$ production, otherwise it might undermine beneficial PTEN functions. Our results comprehensively demonstrated that as an endogenous metabolite, MI enhances mitophagy downstream of PTEN, making it one of the very few endogenous mitophagy activators known[53], and thus open a door

to many potential implications and applications of MI to degenerative diseases where mitophagy is compromised. This is supported by the MI-supplementation mediated decrease of α-SYNUCLEIN puncta in PD models shown here.

Given that MI has been safely used in human as a supplement for a long time, and that our analyses suggest its anti-aging benefits extend from worms to mice through highly a conserved metabolic and signaling pathway present in human, we envision a promising translation potential of MI supplementation

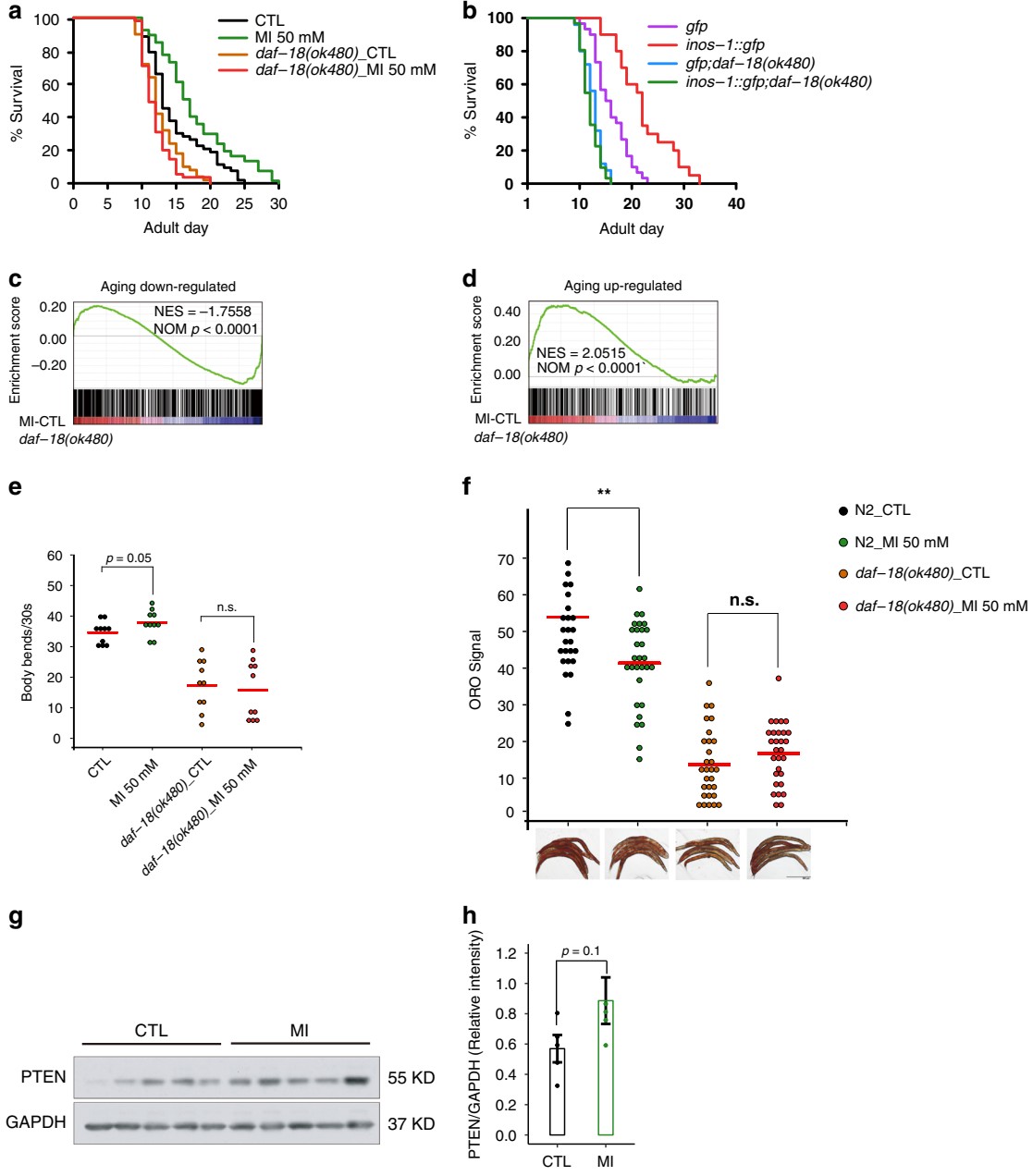

**Fig. 5 The aging-alleviating effect of MI completely depends on DAF-18(PTEN). a** *daf-18*-NL [*daf-18(ok480)*] fully blocked the longevity effect of MI, $N = 4$. **b** *daf-18*-NL fully blocked *inos-1* OE longevity effect, $N = 2$. **c, d** MI supplementation could not reverse the aging-up or downregulated genes expression changes in *daf-18*-NL worms (GSEA $p < 0.0001$ for both), $N = 1$. **e** MI could not increase the mobility of old (AD_8) *daf-18*-NL worms as in control N2 worms ($n = 10$ for each group, two-sided *t*-test n.s. not significant), $N = 2$. **f** MI could not inhibit the fat storage in old (AD_8) *daf-18*-NL worms as in control N2 worms ($n = 30$ in each group, two-sided *t*-test n.s. not significant), $N = 3$. **g** MI supplementation upregulated the PTEN protein level in mouse muscle, $N = 3$. **h** Quantification of the immunoblotting result (two-sided *t*-test $p = 0.1$). Source data are provided as a Source Data file.

to promote healthy human aging (Fig. 4a; Supplementary Figs. 1a, b, 5).

## Methods

**Worm strains and constructs**. Worm strains are listed in Supplementary Table 1. Primers are listed in Supplementary Table 2. Worms were cultured at 20 °C on nematode growth medium (NGM), which were freshly made or air-tightly stored at 4 °C before use.

To create *Psir5::inos-1::GFP::let858.3-3'UTR* construct, a fragment from *inos-1* start to the last amino acid codon was cloned between NotI and XbaI sites of the vector pPD158.87. The empty vector was used as control. Plasmids were injected to young adult worm gonad line to get transgenic worms, without genome integration. Transgenic worms were maintained by picking under fluorescent microscopy.

For RNAi constructs, RNAi sequences were designed using an online tool, https://www.dkfz.de/signaling/e-rnai3/, the fragments were inserted into L4440 vector, between Spe I and Xho I sites. Other RNAi strains are from another library[54].

**Lifespan assay**. Lifespan assay was conducted according to a standard protocol with killed OP50 (Amp supplemented), without FUdR unless otherwise noted[55]. In brief, worms were bleached to get synchronized eggs, these eggs were cultured until breeding, then the naturally bred eggs were picked for lifespan assay. Adult worms of 12 h-after-breeding were deemed as AD_1 and transferred to lifespan assay plates. OP50 were seeded on NGM plates, about 12 h before killing with a Hoefer UVC 520 cross-linker (100 mJ/cm$^2$ for 4 min). Killed OP50 were used to feed worms at least 6 h after the UV procedure. The plates were refreshed daily. Initial about 100 animals were used for each treatment. Dead worms were counted from AD_8. Worms of no response to nose-touching were picked out of the bacteria lawn, then those motionless worms were scored as dead. Worms with an abdomen

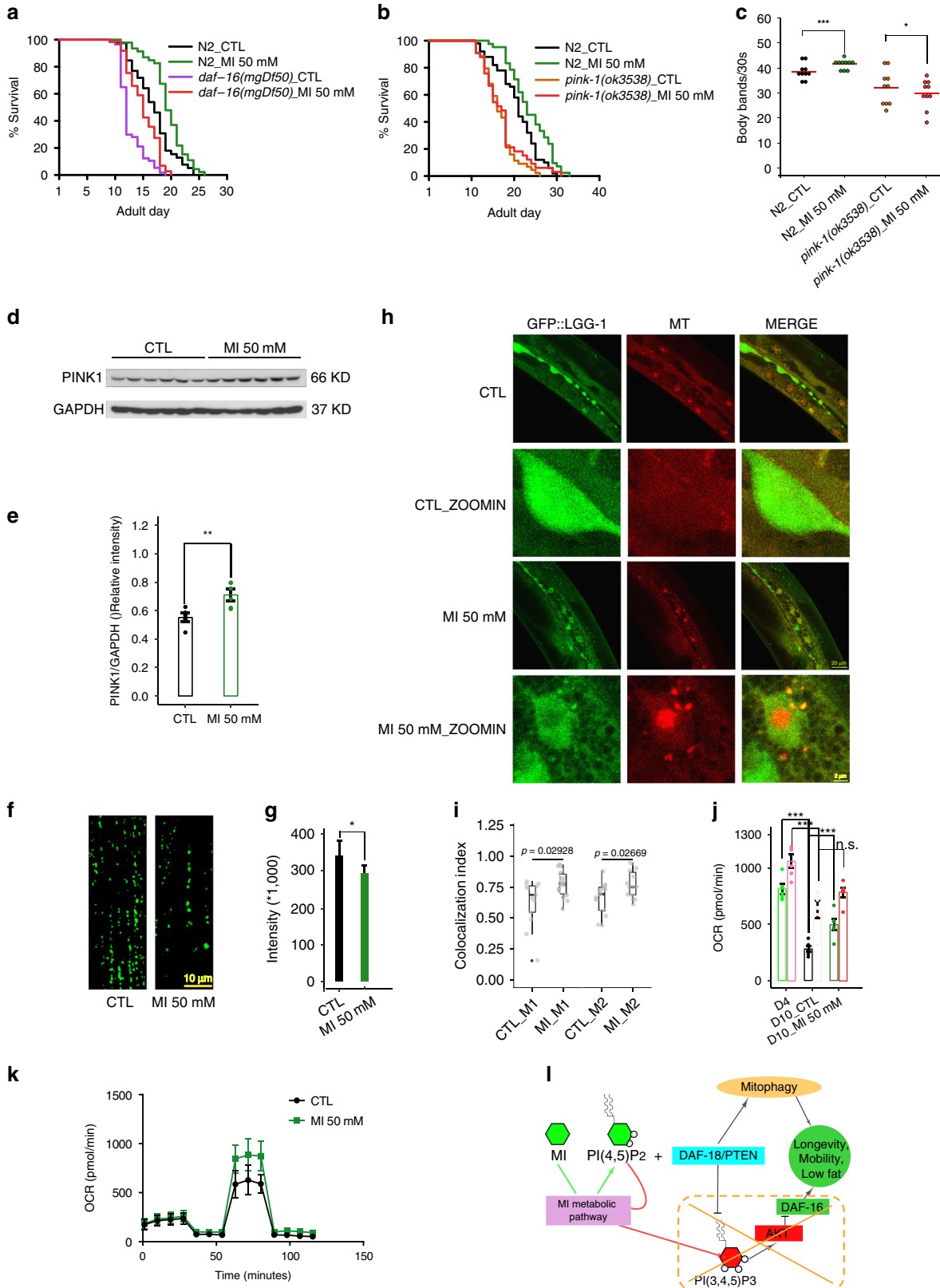

explosion were censored. Statistical analysis of lifespan data is shown in Supplementary Table 3.

**RNA sequencing**. For RNA-seq sample preparation, worms were collected in 1.5 ml tubes, washed several times with M9 to remove bacteria, RNA was extracted with Trizol reagent according to the manufacture's instruction. For mouse samples, equal weight tissues were dissected on ice, collected in 1.5 ml tubes, RNA was extracted

with Trizol reagent. All RNA samples passed the quality control of RIN > 8 and 18S/28S > 1.0. Sequencing was implemented with Illumina HiSeq 2000 system.

**RNA-seq data analysis**. The RNA-seq reads were trimmed and mapped to WS256 genome (from Worm Base) for worm data and mm10 genome (from UCSC genome browser) for mouse data, using STAR (version 2.5)[56] and Rsubread[57]. The read counting was implemented by HTseq using annotations from Worm Base and

**Fig. 6 Mitophagy but not DAF-16 mediates the effects of MI. a** *daf-16*-NL could not block the longevity effect of MI, N = 2. **b** The longevity effect of MI could be largely blocked by *pink-1*-NL (*ok3538*), N = 5. **c** *pink-1*-NL could block the mobility promoting effect of MI seen in control N2 worms (*n* = 10 in each group, \**p* < 0.05), N = 3. **d** Immunoblotting result of PINK1 in differentiated C2C12 cell samples, N = 3. **e** Quantification of the immunoblotting result (Data are presented as mean values ± SEM, two-sided *t*-test \*\**p* < 0.004). **f** Representative images show MI supplementation decreased SQST-1::GFP aggregates in tail muscles of AD_10 worms (*bpIs287[Pmyo-3::sqst-1::gfp]*), scale bar 10 μm, N = 2. **g** Quantification of Fig. 6f like worms, one worm tail region per image, under 63x CONFOCAL microscopy (Data are presented as mean values ± SEM, n = 26, two-sided *t*-test \**p* = 0.005). **h** DA2123 (*adIs2122 [lgg-1p::GFP::lgg-1 + rol-6(su1006)]*) worms were treated for 8 days from AD_1. The figure shows MI induced co-localization of the autophagy marker (GFP::LGG-1) and mitochondria (stained with mito-tracker), scale bar 20 μm for full-size image, scale bar 2 μm for the zoom-in image, N = 2. **i** Quantification of Fig. 6h using the Colocalization Index (Methods). M1 is to quantify the overlapping signal over the mito-tracker signal, and M2 is to quantify the overlapping signal over GFP::LGG-1 signal. MI stands for MI 50 mM. Data are presented as mean values ± SEM, n = 13, p values were determined by a two-sided *t*-test. **j** Mitochondrial respirometry shows MI increased the mitochondrial activity in the worm, left and right represent basal and maximal oxygen respiration rate (OCR) respectively (Data are presented as mean values ± SEM, two-sided *t*-test), N = 2. **k** Mitochondrial respirometry shows MI increased the mitochondrial activity in differentiated C2C12 cells, N = 2. **l** A schematic model summarizes the aging-alleviating effects of MI supplement. The anti-aging effect of MI is through converting to PI(4,5)P₂ and activating PTEN and mitophagy downstream, but not through the classical PTEN downstream insulin IGF-1 pathway components. Source data are provided as a Source Data file.

Gencode, for worm and mouse respectively[58]. The quantification and normalization of reads are conducted by the R package, DESeq2[59]. The term enrichment analyses were conducted with GSEA[60].

**Thrashing assay.** Worms were pretreated as in lifespan assays. On the inspection day, 100 μl M9 buffer was pipetted onto the surface of an empty 35 mm NGM plate, one worm was picked into the buffer, and allowed 30 s to recover from the transfer. Then the number of worm thrashing during 30 s was counted, four times for each worm, 10 to 15 worms for each treatment. One worm from one treatment was counted, then one worm from another treatment, then the third treatment, and so on, the cycle was repeated to avoid the interference of environment or operation.

**Pumping rate assay.** Worms were pretreated as in lifespan assays. All treatment plates were taken out from the incubator to room temperature (RT). The counting was conducted directly on the treatment plates. The number of pumping within 30 s was counted, repeated four times for each worm, 10 to 15 worms per treatment. One worm from one treatment was counted, then one worm from another treatment, then the third treatment, and so on, as thrashing assay. Counted worms were picked out from the culture plates, picked back when the intraday experiments finished.

**ORO staining.** Worms were washed several times with 1× PBS to remove bacteria, then fixed and permeabilized for 30 min by shaking within MRWB buffer containing 2% paraformaldehyde, (2× MRWB buffer: 160 mM KCl, 40 mM NaCl, 14 mM Na₂EGTA, 1 mM spermidine-HCl, 0.4 mM spermine, 30 mM Na-PIPES pH 7.4, 0.2% β-mercaptoethanol). After that, samples were washed twice with 100 mM Tris-Cl (pH 7.4). After wash, the samples were treated with 10 mM DTT in 100 mM Tris-Cl (pH 7.4). Then samples were resuspended in 70% isopropanol in 1× PBS, and incubated for 15 min to dehydrate. Next, the worms were stained with Oil-Red-O solution overnight with shaking (0.5 g/100 ml isopropanol stock solution equilibrated for several days was freshly diluted to 60% with water and rocked for at least 1 h, then filtered with 0.45 μm filter). The next day, the staining solution was discarded and animals were resuspended within pure water. Worms were mounted on 2% agarose pad and imaged under 10× microscope[61]. The intensity was analyzed with an in house MATLAB script.

**Microscopy and mitochondria image analysis.** Worms were anesthetized with 25 mM levamisole, mounted on a 2% agarose pad, with cover clips and fixed with finger nail oil. For NL5901, worm tail region was imaged under 63× CONFOCAL microscopy, one worm per image, 30 animals for each treatment. And the representative picture of five worms was imaged under ×20 objective. For the strain (HZ1770 *[bpIs287(Pmyo-3::sqst-1::gfp)]*), the tail region was imaged under 63× CONFOCAL microscopy. To avoid the influence of levamisole on autophagy, all slides were freshly made and imaged within 5 min. The images were analyzed with the in-house MATLAB scripts.

**Mouse experiment.** The Institutional Animal Care and Use Committee of INS, SINH, CAS, approved all animal procedures and protocols used in this study. The work was performed in adherence to all ethical guidelines. Nine-month-old female C57BL/6J mice were treated with an empty solvent or MI dissolved solution (pH 7.4) by gavage. Drug was given twice weekly, at 580 mg/Kg for an average mouse body weight of 40 g. The food consumption and bodyweight were assessed weekly. After three months, body temperature, serum biochemical indexes, mobility data were acquired. After that, mice were dissected, tissues were instantly frozen in liquid nitrogen, and kept in −80 °C fridge for further use.

**Metabolic pathway.** MI metabolic pathway from KEGG was imported to Cytoscape with the plug-in application KEGGscape[62, 63]. The color of nodes and

edges were labeled according to the RNA expression data and lifespan data in this article.

**ELISA assay.** About 400 worms per sample were homogenized with Bead Rupter 24 Elite (OMNI). The lipid was extracted, and PI(4,5)P₂ quantified according to the kit manual (echelon, K-4500).

**Immunoblotting.** For immunoblotting assay, cells or mouse tissues were homogenized and lysed at 4 °C in lysis buffer (50 mM Tris-HCl, pH 8.0, 1% (v/v) Nonidet P-40, 150 mM NaCl, 5 mM EDTA, 1 mM EGTA, 1 mM sodium orthovanadate, 10 mM sodium fluoride, 1 mM phenylmethylsulfonyl fluoride, 2 μg/ml aprotinin, 5 μg/ml leupeptin, and 1 μg/ml pepstatin). The cell or muscle lysates were centrifuged at 16,000 × *g* for 10 min at 4 °C, and the supernatant was used for immunoblotting analysis. Bio-Rad Protein Assay Dye Reagent was used for measuring protein concentrations in lysates. 20–50 μg of protein was used for immunoblotting. The blotting was conducted following standard protocol.

**Cell culture and oxygen consumption assays.** C2C12 cell line was obtained from the Core Facility for Stem Cell Research in SIBS, and cultured in DMEM with 20% FBS. Cells were seeded on Agilent Seahorse XF24 microplate (Part No. 100777-004), grown to 100% confluency, then 2% horse serum was used to induce differentiation. The differentiated *myo*-tube-like C2C12 was treated with or without 50 mM MI for 24 h in DMEM. Oxygen consumption assay was conducted according to the kit manual, and measured with XF24_3 Extracellular Flux Analyzer. 10 mM Glucose, 1 mM Pyruvtae, and 1 mM GlutaMAX were supplemented in the assay medium. Final 1 μM Oligomycin, 2 μM FCCP, 0.5 μM Rotenoe/Antimycin A were used to interfere with the mitochondria respiration complex. The data were analyzed with Wave (Version -2.6.0.31).

For the oxygen consumption assay of the worm, 40 worms of each treatment, were picked to M9 buffer in XF24 microplate, washed several times to remove bacteria. 10 μM FCCP was used to induce max respiration. Both basal and max respiration were measured six times[64].

**Worm mitochondria staining.** DA2123 worms were treated with MI as lifespan assay until AD_8, then were stained with 0.5 mg/ml Mitotracker (Invitrogen, M7510) in M9 buffer for 3 h. After washing, worms were mounted on a 2% agarose pad, with cover clips and fixed with finger nail oil. Images were made under 63× CONFOCAL microscopy. Manders' coefficient calculated by Coloc2 in Fiji (version 2.0.0) is used as colocalization index in quantification. Both M1 and M2 are calculated.

**Quantification and statistics analysis.** All experiments were repeated at least twice, unless otherwise stated. *n* labels the sample size, which was determined by experience rather than previous statistics, *N* indicates the biological replicates. Student *t*-test was used for significance test, log-rank test was used for survival data analysis. The *p*-values for the GSEA test statistics were calculated by feature permutation for 1000 times.

**Reporting summary.** Further information on research design is available in the Nature Research Reporting Summary linked to this article.

## Data availability

The source data underlying Figs. 1b–d, 2a–f, 3d–f, 4b–f, 5a, b, e–h, 6a–f, g–i and Supplementary Figs. 2a–d, 4a–p, 6a–c, 6f are provided as a Source Data File. Source data are provided with this paper. *C. elegans* and mouse RNA-seq data generated in this study are available at GEO accession number GSE154417. Data can be requested from the corresponding author.

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

## Acknowledgements

We are grateful to Adam Antebi, Yidong Shen, Ye Tian, and Fa-Jun Nan for invaluable advices. We thank the support from I.T. Facility and Uli-Schwarz Central Lab in PICB. This work was supported by grants from the National Natural Science Foundation of China 91749205 and China Ministry of Science and Technology 2016YFE0108700 and Max Planck Fellowship to J.D.J.H.

## Author contributions

J.D.J.H. and D.W.S. conceived the study; J.D.J.H., D.W.S., X.X., and A.Y.C. designed the experiments; D.W.S., A.Y.C., Z.X.X., and J.L. conducted the experiments; D.W.S., J.D.J.H., X.X., Y.Z.Y., G.Y.C., Y.Y.Z., L.H., D.H.C., Y.L., and H.Z. analyzed and interpreted the data; D.W.S., J.D.J.H., X.X., A.Y.C., and J.M.D. wrote the manuscript.

## Competing interests

J.D.J.H. and D.W.S. are inventors on patent application "The anti-aging application of *myo*-inositol" (patent pending # 201910168215.7). The other authors have no competing interests to declare.
