## [Peer Review File · Nature Communications]

Reviewers' comments, first round:

Reviewer #1 (Remarks to the Author):

Dr. Jing-Ding Han's team and colleagues presented a research paper entitled 'The precursor of PI(3,4,5)P3 alleviates aging by activating daf-18(Pten) and independent of daf-16'. By the applications of the using of *C. elegans* and mice, the authors showed that a small natural molecule myo-inositol (MI) exhibits healthy longevity capacity in both worms and mice. Mechanistically, the authors state 'The anti-aging effect of MI is through converting to PI(4,5)P2 and activating PTEN and mitophagy downstream, but not through the classical PTEN downstream insulin IGF-1 pathway components'. The study was very well designed with strong and solid data support the conclusion. The manuscript was relatively well-written with the provisions of sufficient information on statistics, methods, and references.

Addressing some minor questions raised below will help to improve the quality of the paper.

Major concerns:

- Mechanisms: how the MI-daf-18 pathway induces pink1? It would be nice the authors could at least provide additional information based on the available array data they have.
- Other mitophagy pathways. Recent studies show that major mitophagy genes, like pink1, pdr-1 (parkin), and dct-1 play fundamental roles in healthspan (PMID: 30742114; PMID: 25896323). This reviewer wondered whether pdr-1 and dct-1 play any role in MI-induced mitophagy.

Minor concerns:

- For sentence 'such as NAD⁺, spermidine⁵, pyruvate⁶, N-acetylglucosamine (GlcNAc)⁷ and α -ketoglutarate⁸, have been reported to alleviate aging in model organisms'. This reviewer suggests to change 'NAD⁺' to 'NAD⁺ and related metabolites' with additional citations (PMID: 29514064; PMID: 31577933)
- It is suggested to provide more information on the roles of mitophagy in health and ageing (PMID: 30742114; PMID: 30135585 ; PMID: 31375365)
- Does: the authors use 50 mM MI for the worm lifespan and 580 mg/kg for mouse studies. How the authors came out with such doses? Any toxicity experiments?
- Was there any clue that the ARE-Nrf2 (in *C. elegans* the SKN-1 pathway) also plays a role in MI-dependent longevity?

Reviewer #2 (Remarks to the Author):

This referee can comment with some authority on inositol and phosphoinositide function but not on the gene expression studies and their interpretation in this submission. Other than, that is, to observe that the presentation style of much of the manuscript is so obscure and jargon-ridden that only a dedicated expert is likely to understand what is being said. Whatever the decision on its publication fate, I suggest that it is essential that the scientific story it tells must be made much easier for the slightly non-expert to read and understand.

So what is its central story? Essentially that either exposing *Caenorhabditis* worms to lots of inositol or feeding lots of inositol to middle-aged mice appears to improve their healthspan (a word new to me that I liked [until I discovered it is the trade-name of a prominent 'health supplements' company]) – and also to extend lifespan, at least for the worms. These are intriguing observations, particularly in the context of increased recent interest in the possible health benefits of eating extra inositol, particularly in metabolic diseases that include insulin resistance and fatty liver as frequent features (most notably for PCOS and various types of diabetes): for reviews, see Croze & Soulage, 2013, *Biochimie* 95, 1811-1827; Michell, 2018. *Brit J Nutrition*, 120, 301-316. Sadly, there is, as yet, no agreed and coherent explanation for the apparent benefits of eating this cheap and benign dietary supplement.

This manuscript's other main observations, aimed at developing an understanding of these

healthspan/lifespan effects, are of: a) patterns of gene expression changes that might be relevant to aging; and b) the effects of mutationally interfering with metabolic networks in which inositol and its derivatives are (or might be) involved.

The former (see a) above) needed to be clearly presented and interpreted, but were not – at least not to a standard that would be intelligible to anyone other than those few readers who are expert in interpreting transcriptional networks.

The latter (b) pinpointed several enzyme steps that lie on the biosynthetic pathways that lead, amongst other things, to phosphatidylinositol 4,5-bisphosphate (PtdIns(4,5)P₂). The authors therefore suggest that inositol's contribution to the biosynthesis of PtdIns(4,5)P₂ holds the key to its contribution to 'healthspan' and lifespan. However, the authors seem simply to regard PtdIns(4,5)P₂ as a node in a 2-D KEGG network rather than a lipid that has many functions at multiple sites within spatially and temporally complex eukaryotic cells. [For example, what are statements such as "daf-18 . . . is also an enzyme in the MI metabolic pathway" aiming to tell us? Indeed, what do the authors envisage the (singular?) "MI metabolic pathway" to be?].

Given the plethora of functions fulfilled by PtdIns(4,5)P₂ and its downstream metabolites (see, for example, the reviews provided in the MS's refs 12, 14, 21 and 22), it is hardly surprising that interfering with the supply of this pivotal phosphoinositide is prejudicial to long life. It would be amazing if it wasn't – wouldn't it?

Another worrying feature of the presentation is its repeated reference to 'changes' that were statistically 'not significant'. [e.g. p. 4., 6 l. up " . . . also shortened worm lifespan, although non-significantly ($p = 0.737$)"! There is a second such example 4 lines later, and there are others elsewhere (e.g. p. 6 , l. 4 and in the sentence running from p. 7 to p. 8)] When there appears to be a tendency towards change that does not reach the conventional 'statistically significant' probability of $P < 0.05$, it is not normal scientific practice to quote it as a demonstrated change. Otherwise what would be the point of undertaking the statistical analysis at all?

There are a number of other oddities/problems. For example:

p. 4. The first Figure reference in the text is to Fig 4. Why?

p. 3, l. 4. Inositol is not "used as a medication" – a status that requires statutory approval, country by country. As a food supplement it probably does help in some conditions – though the cancer evidence is pretty non-existent. Be precise.

p. 3, midway. "negatively regulates longevity". Meaning "shortens lifespan", I presume.

p. 5. l. 7. Ref 16 is an old review of inositol-deficient fatty liver, but does not justify this general statement made about mammalian body fat.

p.5, midway down. I think there is a "not" missing before M? This whole paragraph is redundant.

p.6, 5 l up. "the lifespan of enzymes"??

Reviewer #3 (Remarks to the Author):

Small molecule metabolites hold great promise as a means to rebalance dysfunctional metabolic pathways that arise during ageing or disease. In this work, Hong and colleagues perform a metabolite screen in *C. elegans* and find that the hexose sugar, myoinositol, can modestly extend motility and life span, reduce neutral lipid accumulation, and ameliorate models of alpha synuclein proteotoxicity (a model of Parkinson's). Similar findings are seen with inos-1 overexpression, which is involved in the production of myoinositol. Both *C. elegans* and cultured cells show transcriptional profiles that suggest changes in age-related genes upon myoinositol supplementation. Furthermore, mice fed myoinositol perform better in open field distance running

and grip strength, suggesting a conserved role in conferring health benefits.

Myoinositol can be diverted to many different molecular species, including phospho-derivatives and phosphatidylinositol lipids. Through genetic experiments, the authors provide suggestive evidence that mutants predicted to accumulate the phosphatidyl inositol lipid PI(4,5)P2 extend life, while those that shunt it away shorten it, implying this species as the active component. Accordingly, myoinositol supplementation results in accumulation of this lipid with ageing. As well, PTEN, which binds to and is activated by this species, is required for life span extension, and somewhat stabilized in cell culture upon supplementation. In contrast DAF-16/FOXO, which mediates the transcriptional output of PTEN and insulin/IGF signaling, is not. Instead, the authors find a role for mitochondrial PINK in life extension, and an effect of myoinositol in stimulating mitochondrial respiration.

This is a very interesting paper that opens up new ideas about the role of myoinositol and derivatives in regulating health and life span. It uses diverse models and means to address this, including *C. elegans*, cell culture and mouse experiments, systems biology and physiology. One of the difficulties of the work that remains, however, is that the effects on life and health span/motility are relatively small, making the epistasis analysis challenging. The life shortening effects of some mutants are not easy to interpret, because of pleiotropic effects on physiology. In addition, the authors measure the internal concentrations of the metabolites in only one case experiment. The authors toggle between different model systems to bolster their arguments, which while appreciated, nevertheless lacks a certain coherence. Nevertheless, I believe the findings are novel and worthy of publication, provided the authors can address the following concerns.

Major

1. The authors make inferences about the levels of metabolites using genetic interventions. In worms, they do measure the levels of PI(4,5)P2 upon myoinositol supplementation during aging using elisa assay. Their case could be strengthened by measuring levels of PI(4,5)P2 upon inos-1 overexpression or in MI treated mice.
2. Along the same lines, the paper might be strengthened if they could show that a known longevity pathway upregulates these metabolites, or the enzymes involved in their production.
3. The authors suggest that the phenotypes are not due to high osmolarity, though 50 mM myoinositol is still considerable. The authors rule out high osmolarity by using mannitol as control throughout, which is appreciated. What happens to myoinositol induced or inos-1 overexpression phenotypes upon knockdown of osmolarity stress mediators such as *gpdh-1*?
4. The authors measure autophagy by measuring levels of p62, and mitophagy by looking co-localization of mitotracker and *lgg-1*. Measuring autophagy is notably, challenging in *C. elegans*, but would at least be strengthened by showing that autophagy or mitophagy gene knockdown reverses some phenotypes. Also what are the controls here? Note that the anesthetic levamisole can significantly affect autophagy even upon short exposure. NaAzide may be a better anaesthetic.
5. It is unclear how many biological replicates were performed for the thrashing assay. If only 10-15 worms were examined per genotype total, and one biological replicate, that is not sufficient.
6. In some cases, the figure quality is poor, particularly 2e, 2f, where the puncta are not visible. Are the numbers of puncta changing, or is the overall intensity? If puncta, then the authors should provide higher magnification and point out the puncta.
7. The paper could be improved with better writing. In many places, the paper is abrupt with little explanation, justification, or rationale given for experiments (see below).
8. Authors should clearly indicate how many times experiments (biological replicates) were done in figure legends. The authors say in methods N= at least 2 for experiments. In my opinion, N should =3, at least for the most critical experiments.

9. I could not find any supplemental tables of strains or aging experiments uploaded. These would be important to evaluate the data.

Minor

10. Pg. 1

The title should include mammalian gene names:

The precursor of PI(3,4,5)P3 alleviates aging by activating daf-18/PTEN and independent of daf-16/FOXO

11. Pg. 4 Correct the word: Results

12. Pg. 4 The authors should report the results of their metabolite screen. What compounds were screened and what were the results? A schematic or graph would be helpful.

13. Fig. 1b, Be consistent about showing/not showing significance of life span extension in figure panel b, c. How many times were these experiments performed, please put in figure legend, mean and max significant?

14. The authors should spell out gene names e.g.

Inos-1= inositol 3 phosphate synthase

ttx-7= inositol monophosphatase

Y6B3B.5=?

15. How many worms were counted for the ORO staining, how many biological replicates?

16. Please correct: We further examined MI's effect on the transcriptome changes.

17. What aging genes are represented under those curves? (please put in supplemental) Were any of the RNA-seq results validated by qPCR?

18. The section on the PC1 and PC2 analysis is confusingly written. What do the authors conclude or interpret from these data? What other biological processes are represented in these principal components. Shouldn't they tie the mitochondria component to examined mitochondrial phenotypes, if that is the point?

19. Page 6 Please explain more clearly what you mean by replicates for all Young CTL, CTL, and MI. Is the pretreatment MI control treated for 9 months, and sacrificed at 9 or 12 months (it should be scored at same age as others, no?)

20. The most significantly up-regulated genes by MI treatment in cultured cells include Ddit4, Npas2, Arrdc3, Foxo1; the most down-regulated genes include Cish, Zfp503 (Fig. S3c). Have any of these been validated by qRT-PCR?

21. PTEN expression level is negatively correlated with BMI in a 280 people human cohort (p=0.017).

Although this reviewer appreciates the human data, it seems to be thrown in as an afterthought, and seems somewhat disconnected from the paper flow. In particular, the potential tie to myoinositol metabolism should be made more clear.

22. Please correct:

We found that MI but not M increased the mobility of Parkinson's disease (PD) model

23. Figure 6i color code is not clear: what is meant by warm colors? Just say right and left represent basal and maximal ocr.

Response to Reviewers

Please note that the reviewers' concerns are listed one by one below (black characters), followed by our responses (blue characters), and when relevant, the corresponding changes in the text (red characters).

Reviewer #1:

Dr. Jing-Ding Han's team and colleagues presented a research paper entitled 'The precursor of PI(3,4,5)P3 alleviates aging by activating daf-18(Pten) and independent of daf-16'. By the applications of the using of *C. elegans* and mice, the authors showed that a small natural molecule myo-inositol (MI) exhibits healthy longevity capacity in both worms and mice. Mechanistically, the authors state 'The anti-aging effect of MI is through converting to PI(4,5)P2 and activating PTEN and mitophagy downstream, but not through the classical PTEN downstream insulin IGF-1 pathway components'. The study was very well designed with strong and solid data support the conclusion. The manuscript was relatively well-written with the provisions of sufficient information on statistics, methods, and references.

Addressing some minor questions raised below will help to improve the quality of the paper.

We thank the reviewer for the precise summary and praise for this manuscript.

Major concerns:

- Mechanisms: how the MI-daf-18 pathway induces pink1? It would be nice the authors could at least provide additional information based on the available array data they have.

Nakamura's lab found *PTEN* overexpression in cancer cells could induce higher expression of *PINK1* at mRNA level¹. We found PTEN protein increased in MI treated mouse muscle (Fig 5g, 5h). We have not observed significant expression level change of *pink-1/Pink1* in MI treatment in either worm or mouse RNA-seq data, however, we did observe PINK1 protein level was up-regulated within MI treated C2C12 cells (Fig. 6d). In mammalian cells there was a report showing that a mitochondria targeting PTEN isoform can bind directly to PINK1 and PARKIN to stabilize the tertiary complex². We therefore tried to detect PINK1 protein level in MI treated mouse muscle, but there are too many PINK1 bands to conclude (some perhaps due to multiple post translational modifications of PINK1 in vivo), hence we opted to leave out this inconclusive data for this study.

- Other mitophagy pathways. Recent studies show that major mitophagy genes, like pink1, pdr-1 (parkin), and dct-1 play fundamental roles in healthspan (PMID: 30742114; PMID: 25896323). This reviewer wondered whether pdr-1 and dct-1 play any role in MI-induced mitophagy.

Loss-of-function of *pdr-1* partially but not as much as *pink-1* blocks MI's lifespan effect (Fig 6c). There is so far no *dct-1* mutant available (perhaps due to its essentiality), we might have to check by RNAi, which is usually conducted with live bacteria, but for MI effects we need to use killed-bacteria, therefore we did not test the effect of *dct-1*.

Minor concerns:

- For sentence 'such as NAD⁺4, spermidine⁵, pyruvate⁶, N-acetylglucosamine (GlcNAc)⁷ and α -ketoglutarate⁸, have been reported to alleviate aging in model organisms'. This reviewer suggests to change 'NAD⁺' to 'NAD⁺ and related metabolites' with additional citations (PMID: 29514064; PMID: 31577933)

This has been modified in manuscript with recommended references added.

- It is suggested to provide more information on the roles of mitophagy in health and ageing (PMID: 30742114; PMID: 30135585 ; PMID: 31375365)

We have now added them, thanks.

- Does: the authors use 50 mM MI for the worm lifespan and 580 mg/kg for mouse studies. How the authors came out with such doses? Any toxicity experiments?

We conducted a dosage assay from 1mM to 700mM MI in worm and used 50mM in the following assays to avoid strong potential osmotic stress (Fig S2d). For mouse experiment, there are many different concentrations in published references³⁻⁵, we used the lowest concentration, 580 mg/kg. We did do toxicity experiments at this and higher concentrations and did not find any liver, kidney or overall toxicity. We have added this explanation to the main text:

“For mouse experiment, we chose the lowest MI concentration reported in the literature³⁻⁵, 580 mg/kg. At this concentration, we did not observe any liver, kidney or overall toxicity.”

- Was there any clue that the ARE-Nrf2 (in *C. elegans* the SKN-1 pathway) also plays a role in MI-dependent longevity?

The transcription factor *skn-1* is essential in regulating the expression of some mitochondria-related genes and mitochondria biogenesis. Loss-of-function of *skn-1* also partially blocked MI lifespan effect as shown below.

Reviewer #2:

This referee can comment with some authority on inositol and phosphoinositide function but not on the gene expression studies and their interpretation in this submission. Other than, that is, to observe that the presentation style of much of the manuscript is so obscure and jargon-ridden that only a dedicated expert is likely to understand what is being said. Whatever the decision on its publication fate, I suggest that it is essential that the scientific story it tells must be made much easier for the slightly non-expert to read and understand.

We thank the reviewer for the suggestions, including those on the writing. We have modified the manuscript accordingly.

So what is its central story? Essentially that either exposing Caenorhabditis worms to lots of inositol or feeding lots of inositol to middle-aged mice appears to improve their healthspan (a word new to me that I liked [until I discovered it is the trade-name of a prominent ‘health supplements’ company]) – and also to extend lifespan, at least for the worms. These are intriguing observations, particularly in the context of increased recent interest in the possible health benefits of eating extra inositol, particularly in metabolic diseases that include insulin resistance and fatty liver as frequent features (most notably for PCOS and various types of diabetes): for reviews, see Croze & Soulage, 2013, Biochimie 95, 1811-1827; Michell, 2018. Brit J Nutrition, 120, 301–316. Sadly, there is, as yet, no agreed and coherent explanation for the apparent benefits of eating this cheap and benign dietary supplement.

We thank the reviewer for his/her recognition on the importance of this study.

This manuscript’s other main observations, aimed at developing an understanding of these healthspan/lifespan effects, are of: a) patterns of gene expression changes that might be relevant to aging; and b) the effects of mutationally interfering with metabolic networks in which inositol and its derivatives are (or might be) involved.

Yes, exactly. we used these methods to discover the underlying mechanism of MI’s aging-alleviating effects in addition to genetic epistasis analysis with other aging regulators.

The former (see a) above) needed to be clearly presented and interpreted, but were not – at least not to a standard that would be intelligible to anyone other than those few readers who are expert in interpreting transcriptional networks.

Thanks for pointing this out. We have replaced the confusing sentences in RNA-seq data analysis paragraph, with the following: “Based on all expressed genes (FPKM>0 in at least 1 sample), PCA of all 9 samples (3 biological replicates each for Young_CTL, CTL and MI) revealed that MI treated muscle samples were closer to CTL samples on PC1 and similar to Young_CTL samples on PC2 (Supplementary Fig. 3a, PC1=46%, PC2=27%). Interestingly, the effect of MI mostly showed on PC2, which is positively related to mitochondrial function as shown by Gene Set Enrichment Analysis (GSEA) (Supplementary Fig. 3a and b). The most significantly up-regulated genes by MI treatment, as compared to CTL samples, include *Ddit4*, *Npas2*, *Arrdc3*, *Foxo1*; the most down-regulated genes include *Cish* and *Zfp503* (Supplementary Fig. 3c).” We also added more description to the legend of Supplementary Fig. 3.

The latter (b) pinpointed several enzyme steps that lie on the biosynthetic pathways that lead, amongst other things, to phosphatidylinositol 4,5-bisphosphate (PtdIns(4,5)P₂). The authors therefore suggest that inositol’s contribution to the biosynthesis of PtdIns(4,5)P₂ holds the key to its contribution to ‘healthspan’ and lifespan. However, the authors seem simply to regard PtdIns(4,5)P₂ as a node in a 2-D KEGG network rather than a lipid that has many functions at multiple sites within spatially and temporally complex eukaryotic cells. [For example, what are statements such as “daf-18 . . . is also an enzyme in the MI metabolic pathway” aiming to tell us? Indeed, what do the authors envisage the (singular?) “MI metabolic pathway” to be?].

We agree with the reviewer, that PtdIns(4,5)P₂ is not only a hub metabolite in MI metabolic pathway, but also a vital lipid, which has many functions at multiple sites within spatially and temporally complex eukaryotic cells. After we found MI’s effect on lifespan, we wondered which downstream metabolite is responsible for its lifespan effect, we searched the MI downstream metabolic reactions as annotated in the “Inositol phosphate metabolism”, “Glycosylphosphatidylinositol (GPI)-anchor biosynthesis”, “Phosphatidylinositol signaling system” KEGG pathways. We have now added this explanation to the text:

“Here we define MI metabolic pathway as the reactions forming a unbroken chain with MI upstream or downstream as annotated by the three KEGG pathways “Inositol phosphate metabolism”, “Phosphatidylinositol signaling system” and “Glycosylphosphatidylinositol (GPI)-anchor biosynthesis”.”

Using genetic perturbations of enzymes shown in these annotated pathways, we found PI(4,5)P₂’s production and consumption in the metabolic network is significantly related to worm lifespan, so is the series of reactions from MI to

PI(4,5)P₂. This suggests that PI(4,5)P₂ regulates worm lifespan, which indeed could involve many biological functions.

In addition, we found that MI induced lifespan extension is fully dependent on *daf-18*, as after knocking out the *daf-18* gene in the worm, MI does not extend worm lifespan. *daf-18* is known to be an enzyme in the metabolic network that uses PI(4,5)P₂ as a activator, which also stimulate the PI(3,4,5)P₃ to PI(4,5)P₂ conversion activity of *daf-18/PTEN*. But *daf-18* also has many other functions independent of its PI(3,4,5)P₃ to PI(4,5)P₂ phosphatase activity, such as protein phosphatase activity and scaffold function⁶.

Given the plethora of functions fulfilled by PtdIns(4,5)P₂ and its downstream metabolites (see, for example, the reviews provided in the MS's refs 12, 14, 21 and 22), it is hardly surprising that interfering with the supply of this pivotal phosphoinositide is prejudicial to long life. It would be amazing if it wasn't – wouldn't it?

Yes, it would not be surprising that blocking its production, or any essential metabolite's production, will decrease lifespan, but that does not predict more of it or any essential metabolite will be beneficial to lifespan. Especially when one downstream metabolite of PI(4,5)P₂ is PI(3,4,5)P₃, of which abnormal higher level is well known to shorten lifespan.

Another worrying feature of the presentation is its repeated reference to 'changes' that were statistically 'not significant'. [e.g. p. 4., 6 l. up “ . . . also shortened worm lifespan, although non-significantly (p = 0.737)”! There is a second such example 4 lines later, and there are others elsewhere (e.g. p. 6 , l. 4 and in the sentence running from p. 7 to p. 8)] When there appears to be a tendency towards change that does not reach the conventional 'statistically significant' probability of P <0.05, it is not normal scientific practice to quote it as a demonstrated change. Otherwise what would be the point of undertaking the statistical analysis at all?

Yes, the statistical *p* value is to confer statistical confidence, but for those that do not reach the statistical *p*<0.05, the trend can be also informative, as we and many others found that they can often be confirmed by additional samples or additional assays using a different method. This is exactly the purpose of exploring these changes for further analysis. Indeed, no conclusion should be made without further confirmation when *p*>0.05. We have removed any assertive claim in such cases.

There are a number of other oddities/problems. For example:
p. 4. The first Figure reference in the text is to Fig 4. Why?

Thanks, we have replaced the reference to Fig 4a by referring to the KEGG pathways containing MI upstream or downstream reaction chains.

p. 3, l. 4. Inositol is not “used as a medication” – a status that requires statutory approval, country by country. As a food supplement it probably does help in some conditions – though the cancer evidence is pretty non-existent. Be precise.

We have changed the expression to “As a common ingredient in food⁷, it is also used as potential medical supplement for diseases⁸, including diabetes⁹ and polycystic ovary syndrome (PCOS)¹⁰.”

p. 3, midway. “negatively regulates longevity”. Meaning “shortens lifespan”, I presume.

We have changed it to “**shortens lifespan**” as suggested by the reviewer.

p. 5. l. 7. Ref 16 is an old review of inositol-deficient fatty liver, but does not justify this general statement made about mammalian body fat.

Thanks, we have now referred to another literature on body fat⁴.

p.5, midway down. I think there is a “not” missing before M? This whole paragraph is redundant.

Thank you for catching this mistake. We have now added “not” before M. It is an important effect of MI to increase mobility of worm Parkinson disease model, which is different from the motility of regular wild type worms in that the Parkinson model use mobility as a readout for alpha-synuclein aggregation and toxicity, whereas the wild type worm motility is a general assessment of muscle functions and coordination. The toxicity induced mobility decline is much earlier, more sudden and severer than the aging dependent motility decline.

p.6, 5 l up. “the lifespan of enzymes”??

We are sorry for this unclarity and thank you for catching it. We have now changed the expression into “To address this, we examined worm lifespans upon perturbations of the enzymes in all the steps of the MI metabolic pathway by using RNAi or LOF mutants (Fig. 4a).”

Reviewer #3:

Small molecule metabolites hold great promise as a means to rebalance dysfunctional metabolic pathways that arise during ageing or disease. In this work, Hong and colleagues perform a metabolite screen in *C. elegans* and find that the hexose sugar, myoinositol, can modestly extend motility and life span, reduce neutral lipid accumulation, and ameliorate models of alpha synuclein proteotoxicity (a model of Parkinson's). Similar findings are seen with inos-1 overexpression, which is involved in the production of myoinositol. Both *C. elegans* and cultured cells show transcriptional profiles that suggest changes in age-related genes upon myoinositol supplementation. Furthermore, mice fed myoinositol perform better in open field distance running and grip strength, suggesting a conserved role in conferring health benefits.

Myoinositol can be diverted to many different molecular species, including phospho-derivatives and phosphatidylinositol lipids. Through genetic experiments, the authors provide suggestive evidence that mutants predicted to accumulate the phosphatidyl inositol lipid PI(4,5)P2 extend life, while those that shunt it away shorten it, implying this species as the active component. Accordingly, myoinositol supplementation results in accumulation of this lipid with ageing. As well, PTEN, which binds to and is activated by this species, is required for life span extension, and somewhat stabilized in cell culture upon supplementation. In contrast DAF-16/FOXO, which mediates the transcriptional output of PTEN and insulin/IGF signaling, is not. Instead, the authors find a role for mitochondrial PINK in life extension, and an effect of myoinositol in stimulating mitochondrial respiration.

This is a very interesting paper that opens up new ideas about the role of myoinositol and derivatives in regulating health and life span. It uses diverse models and means to address this, including *C. elegans*, cell culture and mouse experiments, systems biology and physiology. One of the difficulties of the work that remains, however, is that the effects on life and health span/motility are relatively small, making the epistasis analysis challenging. The life shortening effects of some mutants are not easy to interpret, because of pleiotropic effects on physiology. In addition, the authors measure the internal concentrations of the metabolites in only one case experiment. The authors toggle between different model systems to bolster their arguments, which while appreciated, nevertheless lacks a certain coherence. Nevertheless, I believe the findings are novel and worthy of publication, provided the authors can address the following concerns.

We thank the reviewer for the comprehensive summary of our manuscript.

Major

1. The authors make inferences about the levels of metabolites using genetic interventions. In worms, they do measure the levels of PI(4,5)P2 upon myoinositol supplementation during aging using elisa assay. Their case could be strengthened by measuring levels of PI(4,5)P2 upon inos-1 overexpression or in MI treated mice.

According to the reviewer's suggestion, we have done PI(4,5)P₂ ELISA on mice muscle samples. Compared to Month-3 mice, PI(4,5)P₂ level moderately decreased in Month-12 mice ($p=0.08$), and slightly reversed by MI ($n=3$, $N=3$), but neither of these changes are significant, perhaps due to the large individual variations in mice, which may eventually become significant if we increase the sample size. Unfortunately, we just used up our ELISA kit, and cannot get new ones at the moment by importing from the United States due to the COVID-19 epidemic.

2. Along the same lines, the paper might be strengthened if they could show that a known longevity pathway upregulates these metabolites, or the enzymes involved in their production.

Thanks for the suggestion. As it is not so easy to detect metabolites in the MI pathways, we have examined the enzymes' expression in published microarray or RNA sequencing data, and didn't find a consistent up-regulation of MI or PI(4,5)P₂ producing enzymes by most of the longevity mutants. This is consistent with our epistasis analysis result in worm lifespan, where no complete epistatic relationships were found between MI and *age-1*, *daf-16*, *rsk-1*, *ife-2*, *sir-2.1*, *glp-1*, *isp-1*, *aak-2*, *jnk-1*, *nhr-49*.

3. The authors suggest that the phenotypes are not due to high osmolarity, though 50 mM myoinositol is still considerable. The authors rule out high osmolarity by using mannitol as control throughout, which is appreciated. What happens to myoinositol induced or inos-1 overexpression phenotypes upon knockdown of osmolarity stress mediators such as *gpdh-1*?

According to the reviewer's suggestion, we have added *gpdh-1* result as a confirmation. *gpdh-1* non-allele could not block MI ($N=2$).

We have added this result to the Supplementary Fig. 6f.

4. The authors measure autophagy by measuring levels of p62, and mitophagy by looking co-localization of mitotracker and lgg-1. Measuring autophagy is notably, challenging in *C. elegans*, but would at least be strengthened by showing that autophagy or mitophagy gene knockdown reverses some phenotypes. Also what are the controls here? Note that the anesthetic levamisole can significantly affect autophagy even upon short exposure. NaAzide may be a better anaesthetic.

Yes, for mitophagy genes, we tested *pink-1*(Fig 6b, 6d, 6e) and *pdr-1*(Fig s6d). Loss-of-function of *pink-1* can largely block the lifespan extension by MI, *pdr-1* can partially block.

Also while the manuscript was under review, we tested mitofission *drp-1* and *fis-2* genes by worm mobility assay, which are necessary for mitophagy process¹¹. Both could block MI's mobility enhancement effect (N=2).

We have now added these results to the Supplementary Fig. 6d and described it in the text:

“We further tested mitochondrial fission *drp-1* and *fis-2* genes by worm mobility assay, which are necessary for mitophagy process¹¹. Both could block MI's mobility enhancement effect (N=2).”

Yes, both levamisole and NaAzide could influence autophagy¹², although we used levamisole, we made the slide freshly and instantly imaged to minimize the effect of levamisole, and as negative control, the co-localization was not observed with mannitol treatment and is much less frequent in the vehicle treated blank control.

5. It is unclear how many biological replicates were performed for the thrashing assay. If only 10-15 worms were examined per genotype total, and one biological replicate, that is not sufficient.

We have now described the details in the Methods and biological replicates in figure legend.

The number of worm thrashing in 30 seconds was counted, four times for each worm, 10 to 15 worms for each treatment. We counted 4 times of each worm according to the cited protocol, so it is at least 40 counts for each treatment¹³. We tried to count more worms, however, there would be too many treatments to finish in a single day. And there are more than 2 biological replicates, which are now labeled as N in the manuscript.

6. In some cases, the figure quality is poor, particularly 2e, 2f, where the puncta are not visible. Are the numbers of puncta changing, or is the overall intensity? If puncta, then the authors should provide higher magnification and point out the puncta.

For figure 2e, 2f, worm tail region was imaged under 63X CONFOCAL microscopy, one worm per image, 30 animals for each treatment, the representative picture of 5 worms were imaged under 20X objective. We have now also shown the single worm image at increased resolution, 63x.

We described as “20* labels images under 20x microscopy, 63* for 63x microscopy” in figure legend now.

7. The paper could be improved with better writing. In many places, the paper is abrupt with little explanation, justification, or rationale given for experiments (see below).

We thank the reviewer for the suggestion, we have addressed each of these transitions below as the review pointed out.

8. Authors should clearly indicate how many times experiments (biological replicates) were done in figure legends. The authors say in methods N= at least 2 for experiments. In my opinion, N should =3, at least for the most critical experiments.

Yes, most experiments were repeated more than twice to confirm. We have now specified N in the legend for all the experiments.

9. I could not find any supplemental tables of strains or aging experiments uploaded. These would be important to evaluate the data.

We apologize for missing the supplemental tables in the submission. We have now made sure they are submitted in the revision.

Minor

10. Pg. 1

The title should include mammalian gene names:

The precursor of PI(3,4,5)P3 alleviates aging by activating daf-18/PTEN and independent of daf-16/FOXO

Thanks for the suggestion. But as we only obtained the evidence showing that MI's longevity effect is independent of *daf-16* in worms, but did not do experiment in mice, so to be rigorous, it is better to leave the title unchanged.

11. Pg. 4 Correct the word: Result

We have corrected this in the manuscript.

12. Pg. 4 The authors should report the results of their metabolite screen. What compounds were screened and what were the results? A schematic or graph would be helpful.

We are continuing the systematic screen on worm metabolites, which will be reported in another study. Our initial partial screen did not have enough biological replicates and did not have the intended systematic coverage of metabolic pathways, we do not want to distribute potentially unreliable information and will provide the full list in the next paper that comprehensively describe the screen results.

13. Fig. 1b, Be consistent about showing/not showing significance of life span extension in figure panel b, c. How many times were these experiments performed, please put in figure legend, mean and max significant?

We have now summarized all the lifespan experiment and statistics in Supplementary Table3.

14. The authors should spell out gene names e.g.
Inos-1= inositol 3 phosphate synthase
ttx-7= inositol monophosphatase
Y6B3B.5=?

We have added the full gene names in abbreviation. For some genes there are no gene names except clone ID, such as “Y6B3B.5”.

15. How many worms were counted for the ORO staining, how many biological replicates?

We used 30 worms for each condition (one worm per image), and 6 biological replicates for N2 worms (2 for Fig 2c, 2 for Fig 5f), 3 biological replicates for *daf-18(ok480)* worms.

16. Please correct: We further examined MI's effect on the transcriptome changes.

Thanks, we have corrected it the manuscript.

17. What aging genes are represented under those curves? (please put in supplemental) Were any of the RNA-seq results validated by qPCR?

The aging gene lists are now added to the manuscript as supplementary excel file, the new Supplemental Dataset 1. As there is a good consistency among replicates, and other aging transcriptomes, we didn't do validation by qPCR.

18. The section on the PC1 and PC2 analysis is confusingly written. What do the authors conclude or interpret from these data? What other biological processes are represented in these principal components. Shouldn't they tie the mitochondria component to examined mitochondrial phenotypes, if that is the point?

Thanks for pointing this out. We have replaced the confusing sentences in RNA-seq data analysis paragraph, with the following: “Based on all expressed genes (FPKM>0 in at least 1 sample), PCA of all 9 samples (3 biological replicates each for Young_CTL, CTL and MI) revealed that MI treated muscle samples were closer to CTL samples on PC1 and similar to Young_CTL samples on PC2 (Supplementary Fig. 3a, PC1=46%, PC2=27%). Interestingly, the effect of MI mostly showed on PC2, which is positively related to mitochondrial function as shown by Gene Set Enrichment Analysis (GSEA) (Supplementary Fig. 3a and b). The most significantly up-regulated genes by MI treatment, as compared to CTL samples, include *Ddit4*, *Npas2*, *Arrdc3*, *Foxo1*; the most down-regulated genes include *Cish* and *Zfp503* (Supplementary Fig. 3c).” We also added more description to the legend of Supplementary Fig. 3.

19. Page 6 Please explain more clearly what you mean by replicates for all Young CTL, CTL, and MI. Is the pretreatment MI control treated for 9 months, and sacrificed at 9 or 12 months (it should be scored at same age as others, no?)

To minimize technical variations, all mice were scored and sacrificed at the same time “CTL” and “MI” are of already 12 months, “Pre-treatment CTL” and “Young CTL” are 9 months and 3 months, respectively. We have illustrated the experimental design in Fig 3a.

20. The most significantly up-regulated genes by MI treatment in cultured cells include *Ddit4*, *Npas2*, *Arrdc3*, *Foxo1*; the most down-regulated genes include *Cish*, *Zfp503* (Fig. S3c). Have any of these been validated by qRT-PCR?

As there is a good consistency among RNA-seq replicates for these genes (Fig. S3c), we did not perform qRT-PCR.

21. PTEN expression level is negatively correlated with BMI in a 280 people human cohort ($p=0.017$). Although this reviewer appreciates the human data, it seems to be thrown in as an afterthought, and seems somewhat disconnected from the paper flow. In particular, the potential tie to myoinositol metabolism should be made more clear.

According to the reviewer’s suggestion, we have removed this data from the manuscript.

22. Please correct:

We found that MI but not M increased the mobility of Parkinson’s disease (PD) model

Corrected, thanks.

23. Figure 6i color code is not clear: what is meant by warm colors? Just say right and left represent basal and maximal ocr.

We have changed the figure and legend accordingly.

Reference:

- 1 Unoki, M. & Nakamura, Y. Growth-suppressive effects of BPOZ and EGR2, two genes involved in the PTEN signaling pathway. *Oncogene* **20**, 4457-4465, doi:10.1038/sj.onc.1204608 (2001).
- 2 Liang, H. *et al.* PTEN α , a PTEN Isoform Translated through Alternative Initiation, Regulates Mitochondrial Function and Energy Metabolism. *Cell Metab* **19**, 836-848, doi:<http://dx.doi.org/10.1016/j.cmet.2014.03.023> (2014).
- 3 Croze, M. L., Geloën, A. & Soulage, C. O. Abnormalities in myo-inositol metabolism associated with type 2 diabetes in mice fed a high-fat diet: benefits of a dietary myo-inositol supplementation. *Br J Nutr* **113**, 1862-1875, doi:10.1017/s000711451500121x (2015).
- 4 Croze, M. L. *et al.* Chronic treatment with myo-inositol reduces white adipose tissue accretion and improves insulin sensitivity in female mice. *J Nutr Biochem* **24**, 457-466, doi:10.1016/j.jnutbio.2012.01.008 (2013).
- 5 Dang, N. T., Mukai, R., Yoshida, K. & Ashida, H. D-pinitol and myo-inositol stimulate translocation of glucose transporter 4 in skeletal muscle of C57BL/6 mice. *Bioscience, biotechnology, and biochemistry* **74**, 1062-1067, doi:10.1271/bbb.90963 (2010).
- 6 Lee, Y. R., Chen, M. & Pandolfi, P. P. The functions and regulation of the PTEN tumour suppressor: new modes and prospects. *Nat Rev Mol Cell Biol*, doi:10.1038/s41580-018-0015-0 (2018).
- 7 Clements, R. S. & Darnell, B. Myoinositol Content of Common Foods - Development of a High-Myo-Inositol Diet. *American Journal of Clinical Nutrition* **33**, 1954-1967 (1980).
- 8 Holub, B. J. Metabolism and function of myo-inositol and inositol phospholipids. *Annu Rev Nutr* **6**, 563-597, doi:10.1146/annurev.nu.06.070186.003023 (1986).
- 9 Santamaria, A. *et al.* One-year effects of myo-inositol supplementation in postmenopausal women with metabolic syndrome. *Climacteric : the journal of the International Menopause Society* **15**, 490-495, doi:10.3109/13697137.2011.631063 (2012).
- 10 Unfer, V., Nestler, J. E., Kamenov, Z. A., Prapas, N. & Facchinetti, F. Effects of Inositol(s) in Women with PCOS: A Systematic Review of Randomized Controlled Trials. *International journal of endocrinology* **2016**, 1849162, doi:10.1155/2016/1849162 (2016).
- 11 van der Bliek, A. M., Sedensky, M. M. & Morgan, P. G. Cell Biology of the Mitochondrion. *Genetics* **207**, 843-871, doi:10.1534/genetics.117.300262 (2017).
- 12 Zhang, H. *et al.* Guidelines for monitoring autophagy in *Caenorhabditis elegans*. *Autophagy* **11**, 9-27, doi:10.1080/15548627.2014.1003478 (2015).
- 13 Keith, S. A., Amrit, F. R., Ratnappan, R. & Ghazi, A. The *C. elegans* healthspan and stress-resistance assay toolkit. *Methods* **68**, 476-486, doi:10.1016/j.ymeth.2014.04.003 (2014).

Reviewers' comments, second round:

Reviewer #1 (Remarks to the Author):

Many key experiments were performed by the authors and with the data the authors have convincingly addressed the majority of the questions raised by this reviewers and the other reviewers(s). The conclusions of the current paper are solid.

Reviewer #2 (Remarks to the Author):

This revised manuscript is considerably improved. however, there are two points which I would like to see clarified before acceptance.

A point that I made previously (as reviewer #2) was that the authors repeatedly offered as evidence, and then interpreted, experimental 'results' that failed to reach statistical significance.

The authors accepted this criticism, but offered some less than convincing justification for inclusion of this equivocal information and their interpretations thereof. And they concluded their response to the criticism with "We have removed any assertive claim in such cases."

BUT they haven't. At lines 232-3, they still baldly state that "We found that PTEN protein was OBVIOUSLY but not significantly up regulated in muscle ($P=0.1$)." What sort of scientific argument is this?

A second, more minor, point on l. 49. Why does PtdIns4P have a redundant (and incorrect) 1- in front of it?

Reviewer #3 (Remarks to the Author):

The authors have mostly addressed my concerns.

A few small points:

1. Insert "it":

we used 50 mM mannitol (M)¹⁰⁹ as an osmotic control, because cannot be it further metabolized in worms.

2. Although MI could treat polycystic ovary syndrome (PCOS), it did not obviously affect 135 worm brood size when treating the worms from L4 stage to AD₈ (Supplementary Fig. 2c).

Unclear what does PCOS have to do with brood size?

3. insert "the":

which are necessary for the mitophagy process.

4. For the colocalization experiments, they should quantitative the amount of colocalization e.g. with Manders coefficient.

5. Did the authors ever try to supplement myoinositol to mutants deficient in its production to see if it rescues longevity? At this point I don't think it necessary, but if the experiments have been done, it might solidify the results.

Response to Reviewers

Please note that the reviewers' concerns are listed one by one below (black characters), followed by our responses (blue characters), and when relevant, the corresponding changes in the text (red characters).

Reviewer #1:

Many key experiments were performed by the authors and with the data the authors have convincingly addressed the majority of the questions raised by this reviewers and the other reviewers(s). The conclusions of the current paper are solid.

We thank reviewer #1 for the confirmation of the improvement.

Reviewer #2 (Remarks to the Author):

This revised manuscript is considerably improved. however, there are two points which I would like to see clarified before acceptance.

- A point that I made previously (as reviewer #2) was that the authors repeatedly offered as evidence, and then interpreted, experimental 'results' that failed to reach statistical significance.

The authors accepted this criticism, but offered some less than convincing justification for inclusion of this equivocal information and their interpretations thereof. And they concluded their response to the criticism with "We have removed any assertive claim in such cases."

BUT they haven't. At lines 232-3, they still baldly state that "We found that PTEN protein was OBVIOUSLY but not significantly up regulated in muscle ($P=0.1$)." What sort of scientific argument is this?

We thank the reviewer for the precise statement. And we changed the description as suggested to:

"We found that mouse PTEN protein was mildly but not significantly up-regulated in muscle ($p=0.1$)."

- A second, more minor, point on l. 49. Why does PtdIns4P have a redundant (and incorrect) 1- in front of it?

For the abbreviation of *ppk-1*, its full name is "1-phosphatidylinositol-4-phosphate 5-kinase", as annotated in KEGG (https://www.kegg.jp/dbget-bin/www_bget?cel:CELE_F55A12.3+cel:CELE_VF11C1L.1). Indeed, '1-' is redundant for *ppk-2* and *piki-1*. We have now removed it. We thank the reviewer for catching it.

Reviewer #3 (Remarks to the Author):

The authors have mostly addressed my concerns.

A few small points:

1. Insert "it":

we used 50 mM mannitol (M)109 as an osmotic control, because cannot be it further metabolized in worms.

Thanks for catching this typo, we have corrected this to “because it cannot be further metabolized in worms”.

2. Although MI could treat polycystic ovary syndrome (PCOS), it did not obviously affect 135 worm brood size when treating the worms from L4 stage to AD_8 (Supplementary Fig. 2c).

Unclear what does PCOS have to do with brood size?

MI was used for the treatment of PCOS, which leads to female infertility, with unknown mechanism. Although MI metabolism is conserved from worm to human, MI did not influence worm brood size. We thank the reviewer for the remark, and changed the expression as:

“We also checked the effect of MI on worm reproduction, and found that it did not obviously affect worm brood size when treating the worms from L4 stage to AD_8”

3. insert "the":

which are necessary for the mitophagy process.

We have corrected this, thanks.

4. For the colocalization experiments, they should quantitative the amount of colocalization e.g. with Manders coefficient.

Thanks for the comment. We have done this as suggested, which is shown below and added to the manuscript.

5. Did the authors ever try to supplement myoinositol to mutants deficient in its production to see if it rescues longevity? At this point I don't think it necessary, but if the experiments have been done, it might solidify the results.

We thank the reviewer for the reviewer's remark. Yes, such an experiment could further solidify the result. There are two steps to synthesize MI, with *inos-1* for the first step, *txx-7* for the second. It is possible to do the experiment by feeding *txx-7*-LOF mutant worms with MI. However, as there are already 3 experiments to confirm the longevity effect of MI (Fig. 1b – 1d), which are stronger evidence than a rescue experiment, we did not do this rescue experiment.